# From coarse to fine: The absolute *Escherichia coli* proteome under diverse growth conditions

Matteo Mori[1] [ID], Zhongge Zhang[2], Amir Banaei-Esfahani[3] [ID], Jean-Benoît Lalanne[4,5] [ID], Hiroyuki Okano[1], Ben C Collins[3,6] [ID], Alexander Schmidt[7] [ID], Olga T Schubert[8] [ID], Deok-Sun Lee[9], Gene-Wei Li[4] [ID], Ruedi Aebersold[3,10,*,†] [ID], Terence Hwa[1,2,**,†] [ID] & Christina Ludwig[11,***,†] [ID]

## Abstract

Accurate measurements of cellular protein concentrations are invaluable to quantitative studies of gene expression and physiology in living cells. Here, we developed a versatile mass spectrometric workflow based on data-independent acquisition proteomics (DIA/SWATH) together with a novel protein inference algorithm (xTop). We used this workflow to accurately quantify absolute protein abundances in *Escherichia coli* for > 2,000 proteins over > 60 growth conditions, including nutrient limitations, non-metabolic stresses, and non-planktonic states. The resulting high-quality dataset of protein mass fractions allowed us to characterize proteome responses from a coarse (groups of related proteins) to a fine (individual) protein level. Hereby, a plethora of novel biological findings could be elucidated, including the generic upregulation of low-abundant proteins under various metabolic limitations, the non-specificity of catabolic enzymes upregulated under carbon limitation, the lack of large-scale proteome reallocation under stress compared to nutrient limitations, as well as surprising strain-dependent effects important for biofilm formation. These results present valuable resources for the systems biology community and can be used for future multi-omics studies of gene regulation and metabolic control in *E. coli*.

**Keywords** absolute quantification; *Escherichia coli*; mass spectrometry; protein inference; quantitative proteomics
**Subject Categories** Metabolism; Microbiology, Virology & Host Pathogen Interaction; Proteomics

**Mol Syst Biol. (2021) 17: e9536**

## Introduction

Proteins are one of the key molecular players in living cells, directly affecting cell behavior through myriads of activities. They are controlled, regulated, and fine-tuned in time and space through various mechanisms, including protein synthesis, turnover, post-translational modifications, and protein–protein interactions. In comparison to DNA or RNA, proteins represent a more direct read-out of cellular functions and phenotypes, since proteins are the biomolecules catalyzing most biochemical reactions. Therefore, quantitative measurements of proteins, their turnover rates, their modification, or their interaction status provide direct snapshots of cellular processes, allowing to associate gene expression to physiology and phenotypes. Over the last decades, liquid-chromatography coupled to tandem mass spectrometry (LC-MS/MS) has matured to be the method of choice for generating quantitative proteomic data (Aebersold & Mann, 2003, 2016). A specific challenge for systems-level studies is the reliable quantification of thousands of proteins, including proteins at low concentrations, across large sample cohorts from a variety of different growth conditions, phenotypes, or strains (Rost *et al*, 2015). Both relative protein quantification (allowing cross-sample comparisons for the same protein) and absolute protein quantification (allowing cross-protein comparisons in the same sample) provide crucial information on the activity of

1  Department of Physics, University of California at San Diego, La Jolla, CA, USA
2  Section of Molecular Biology, Division of Biological Sciences, University of California at San Diego, La Jolla, CA, USA
3  Department of Biology, Institute of Molecular Systems Biology, ETH Zurich, Zurich, Switzerland
4  Department of Biology, Massachusetts Institute of Technology, Cambridge, MA, USA
5  Department of Physics, Massachusetts Institute of Technology, Cambridge, MA, USA
6  School of Biological Sciences, Queen's University of Belfast, Belfast, UK
7  Biozentrum, University of Basel, Basel, Switzerland
8  Department of Human Genetics, University of California, Los Angeles, Los Angeles, CA, USA
9  School of Computational Sciences, Korea Institute for Advanced Study, Seoul, Korea
10 Faculty of Science, University of Zurich, Zurich, Switzerland
11 Bavarian Center for Biomolecular Mass Spectrometry (BayBioMS), Technical University of Munich (TUM), Freising, Germany
   *Corresponding author. Tel: +41 44 633 3170; E-mail: aebersold@imsb.biol.ethz.ch
   **Corresponding author. Tel: +1 858 534 7263; E-mail: hwa@ucsd.edu
   ***Corresponding author. Tel: +49 8161 71 6199; E-mail: tina.ludwig@tum.de
   †These authors contributed equally to this work

biochemical and regulation pathways, the stoichiometry of protein complexes, and the relationship between gene expression and cellular phenotype (Ludwig & Aebersold, 2014; Schubert *et al*, 2015; Schmidt *et al*, 2016). Furthermore, accurate measurements of absolute protein abundances (e.g., "number of proteins per cell" or "protein mass fractions"), together with the knowledge of cell volume, give the cellular protein concentrations (Appendix Note S1). This can be combined with other omics data to yield detailed biochemical information. For example, translational efficiencies of mRNA can be obtained if data on absolute mRNA concentrations are available (Li *et al*, 2014), or enzymatic parameters can be investigated if concentrations of metabolites associated with an enzyme and the flux it carries are known (Schubert *et al*, 2015).

The Gram-negative bacterium *Escherichia coli* is one of the best-characterized model organism, and a workhorse for microbial genetics, biotechnology, and systems biology, thanks to many decades of rigorous molecular and physiological studies (Lee, 1996; Neidhardt, 1996; Bremer & Dennis, 2008; Karp *et al*, 2018). In the past decade, substantial advancements have been made in the quantitative characterization of the proteome of *E. coli*, driven in part by elucidating the cost of protein synthesis and the allocation of proteomic resources in different growth conditions (Basan *et al*, 2015a; Hui *et al*, 2015; Peebo *et al*, 2015; Caglar *et al*, 2017; Erickson *et al*, 2017). Most of these proteomic studies focused on the absolute abundances of groups of proteins, e.g., the abundances of all enzymes involved in glycolysis or in amino acid synthesis. Quantitative data on protein abundances collected at this coarse-grained level across a spectrum of relevant growth conditions showed that the cost of protein synthesis is key to explain a number of ubiquitous microbial phenomena, e.g., catabolite repression (You *et al*, 2013; Hui *et al*, 2015), metabolic overflow (Basan *et al*, 2015a; Peebo *et al*, 2015), and diauxic shift (Erickson *et al*, 2017). While the accuracy of quantitative proteomics at that time was not sufficient for making quantitative statements on the abundances of individual proteins, abundance estimates based on ribosome profiling were able to generate insightful quantitative information at the individual protein level, e.g., in quantitatively assessing the fitness effect of the expression of a single metabolic protein and on the stoichiometric relation between enzymes in protein complexes (Li *et al*, 2014). However, the elaborate workflow and high costs of ribosome profiling make this demanding method difficult to apply to a large number of growth conditions.

A large step in the direction of comprehensive quantitation of *E. coli* proteomes was made by Schmidt *et al* (2016), who calibrated mass spectrometric protein intensities using quantified external standards (AQUA peptides) for a subset of 41 proteins expressed at different abundances. This study investigated proteome allocation, expression regulation, and post-translational adaptations of *E. coli* across a set of 22 different growth conditions. However, despite the improvement in quantitation, their major findings either only considered the total abundance of groups of proteins or were not quantitative in nature. A detailed analysis presented in this work showed that in fact the accuracy of absolute abundance quantitation using AQUA peptide calibration is limited. One key challenge for accurate quantitation of absolute protein abundances in bottom-up proteomics is that peptides, rather than proteins, are the measured analytes. Therefore, absolute protein abundance needs to be inferred from peptide abundances, which is not straightforward—

different peptide precursors from the same protein can yield very different intensities. Even when external standards, such as AQUA peptides, are used, they provide only information on proteins from which the peptides are derived from. Additionally, accurate absolute quantification with AQUA peptides is very expensive, work intensive, technically challenging, and can still be error-prone (Ludwig & Aebersold, 2014).

In this study, we described a versatile workflow that accurately quantifies absolute abundances of thousands of *E. coli* proteins at the individual protein level over many conditions. We demonstrated the usefulness of the generated datasets by providing extensive biological analyses of numerous individual proteins, which is something that has not been done previously in proteomic studies of *E. coli*. Additional utility at the individual protein level will be shown in follow-up studies, where we will combine the data generated here with other omics approaches. Compared to previous studies, our approach provides high-throughput quantification that is comprehensive, accurate, and reproducible, and delivers at low costs and a reasonably fast timescale (1 h per sample). Our pipeline is based on data-independent acquisition mass spectrometry (DIA/SWATH (Gillet *et al*, 2012; Chapman *et al*, 2014; Ludwig *et al*, 2018)) for which we generated a tailor-made comprehensive *E. coli* spectral library entailing information for 64 % of all annotated *E. coli* proteins. DIA/SWATH mass spectrometry applied to study *E. coli* proteomes has recently been shown to provide excellent quantitative results in terms of precision, reproducibility, and deep proteome coverage (Midha *et al*, 2020).

Further, we established a novel peptide-to-protein inference algorithm, named xTop, which combines intensities from unique peptides of a given protein across *all* samples at hand to infer the intensities of that protein. We showed that xTop is superior in estimating relative protein abundances across samples, compared to other commonly used algorithms, such as iBAQ (Schwanhausser *et al*, 2011) or TopPepN (Silva *et al*, 2006; Ludwig *et al*, 2012; Rosenberger *et al*, 2014). We benchmarked these protein inference methods, along with ribosome profiling, for their estimate of absolute protein abundances against a set of spiked-in reference peptides (AQUA), as well as by using a number of internal references offered by protein complexes with known stoichiometry. We established that absolute protein abundances inferred from ribosome profiling data are superior in accuracy. We therefore calibrated the relative protein abundances provided by proteomics and xTop to the absolute abundance obtained from ribosome profiling, hence obtaining accurate protein abundances across a vast number of samples.

Finally, we applied our workflow to explore the *E. coli* proteome across ~ 60 growth conditions. Here we extended well beyond nutrient limitation (carbon, phosphate, oxygen) and included anaerobic growth, various non-metabolic stresses (high temperature, hyperosmolarity, acetate, ethanol, oxidative), and conditions favoring non-planktonic growth, such as biofilm and colony growth. A total of 2,335 proteins were detected from 66 samples across these conditions. This comprehensive dataset allowed us to characterize proteome responses at the global level and, crucially, for individual proteins at unprecedented detail. At the level of protein sectors (groups of proteins exhibiting similar response patterns under metabolic limitations or antibiotic inhibition), the responses by abundant proteins were found to match what was previously seen for sector aggregates (Peebo *et al*, 2015; Schmidt *et al*, 2016; Caglar *et al*,

2017). However, a large number of newly detected, low-abundant proteins exhibited distinct responses unresolved in previous studies. A more detailed examination of individual proteins in nutrient limitation, stress conditions, and for various commonly used media and genotypes revealed several surprises, including the commonality of the response to growth on different carbon sources, the impact of micronutrients in growth medium, the lack of proteome-wide response to non-metabolic stresses, and factors affecting motility and biofilm formation. These findings shed new light on physiological responses of *E. coli* to environmental and genetic perturbations, and generate a variety of interesting hypotheses to be further examined by follow-up studies.

# Results

### Workflow development

We developed a versatile workflow for relative and absolute quantification of *E. coli* proteomes across many samples using DIA/SWATH mass spectrometry. For the peptide-centric analysis of DIA/SWATH data, a "spectral library" encapsulating prior knowledge about chromatographic and mass spectrometric behavior of peptides is required. We generated a comprehensive *E. coli* spectral library from a diverse set of *E. coli* proteomes. Further, we developed a novel protein inference algorithm, termed xTop, and tested its performance in comparison to other commonly used inference algorithms, such as iBAQ (Schwanhausser *et al*, 2011) and TopPepN (Silva *et al*, 2006; Ludwig *et al*, 2012; Rosenberger *et al*, 2014).

### Spectral library generation

To generate a comprehensive *E. coli* spectral library for peptide-centric DIA/SWATH data analysis, we followed the workflow illustrated in Fig 1A. To detect as many peptides and proteins as possible, including those proteins that are expressed only under specific growth conditions, we grew *E. coli* cells in 34 diverse growth conditions, including exponential, stationary, and biofilm-forming conditions, exposure to a spectrum of stresses (high and low pH, hyperosmolarity, high temperature, oxidative stress), as well as a wide range of nutrient sources (Datasets EV1 and EV2). All 34 samples were measured by DDA-based mass spectrometry on a quadrupole-time-of-flight mass spectrometer (TripleTOF 5600, Sciex). To further increase proteome coverage, a pooled sample was fractionated by peptide off-gel electrophoresis (OGE) into 13 fractions, which were measured individually by DDA proteomics. This approach allowed us to increase the peptide coverage from ~ 10,000 for a typical DDA measurement to a total of 26,285 unique peptide sequences, corresponding to 2,770 unique *E. coli* proteins (64% of all annotated *E. coli* proteins) (Fig 1B). About ¾ of the identified proteins have been detected with more than three peptides (Fig 1C). The resulting spectral library is freely available through the SWATHAtlas repository in different formats (PASS01421) and can be used by the mass spectrometric community as a comprehensive resource for acquiring and analyzing mass spectreometic data from the model organism *E. coli*.

### From peptides to proteins: the xTop algorithm

Next, we developed a novel quantitative protein inference algorithm, termed "xTop", which exploits and combines information

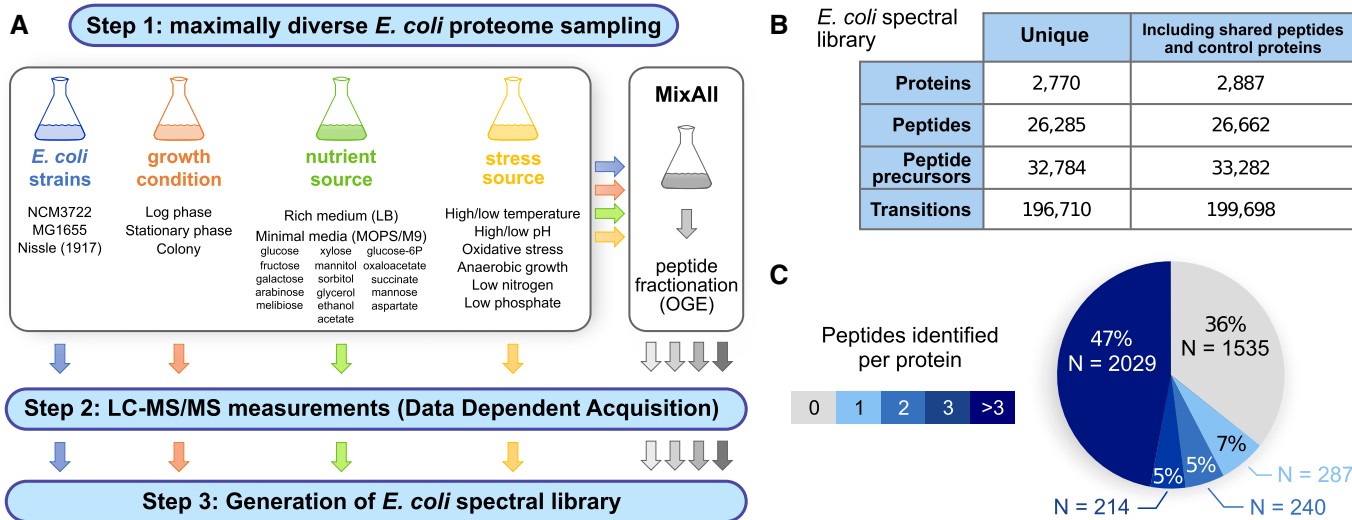

**Figure 1. Spectral library generation to target the *Escherichia coli* proteome.**

A Workflow employed to generate a comprehensive *E. coli* spectral library. Step 1: A wide range of *E. coli* cells from various strains grown under different conditions were generated, including different time points of sampling, growth media, high and low pH, aerobic and anaerobic growth, temperatures, high and low osmotic conditions, and different nutrition additives. Peptide fractionation by off-gel electrophoresis (OGE) was performed on a "MixAll" sample. Step 2: All samples were measured in data-dependent acquisition (DDA) mode on a TripleTOF 5600 instrument. In total, 53 MS injections were performed. Step 3: MS2 spectra were matched to the canonical *E. coli* proteome, and a consensus spectral library was generated.

B Numbers of proteins, peptides, precursors, and transitions entailed in the *E. coli* spectral library. Given are the statistics for the unique proteins and peptides only, as well as for all entries, including also shared peptides, iRT peptides as well as 9 control proteins not from the organism *E. coli*.

C Distribution of detectable unique peptides per protein.

from *all* peptides of a given protein detected across *all* samples to infer the absolute protein intensity in each sample. Salient features of the xTop algorithm are illustrated in Fig 2A. For each protein, the intensities of its peptide precursors $p$ across each sample $s$ are represented as a matrix element $I_{ps}$. This matrix is modeled as the product of two components, the sample-dependent xTop intensity $I_s^{\mathrm{xTop}}$, and the peptide-specific detection efficiency $\epsilon_p$. These two components are determined from the data matrix $I_{ps}$ from their *maximum a posteriori probability* (MAP) estimators (summarized in Figure N2.2 within Appendix Note S2). Importantly, the xTop protein intensity is obtained as a weighted average of all peptide

precursors intensities. Peptides whose intensities display a large degree of mutual consistency across samples contribute the most to the intensity $I_s^{\mathrm{xTop}}$, while peptides weakly correlated with the others contribute the least. Therefore, this method mitigates the impact of missing or noisy peptide precursors on the inferred protein intensities. An in-depth description of the method and of its implementation is provided in Appendix Note S2.

### Assessment of xTop performance

In order to validate the xTop method and benchmark it against various other commonly used protein inference methods (TopPep1/3

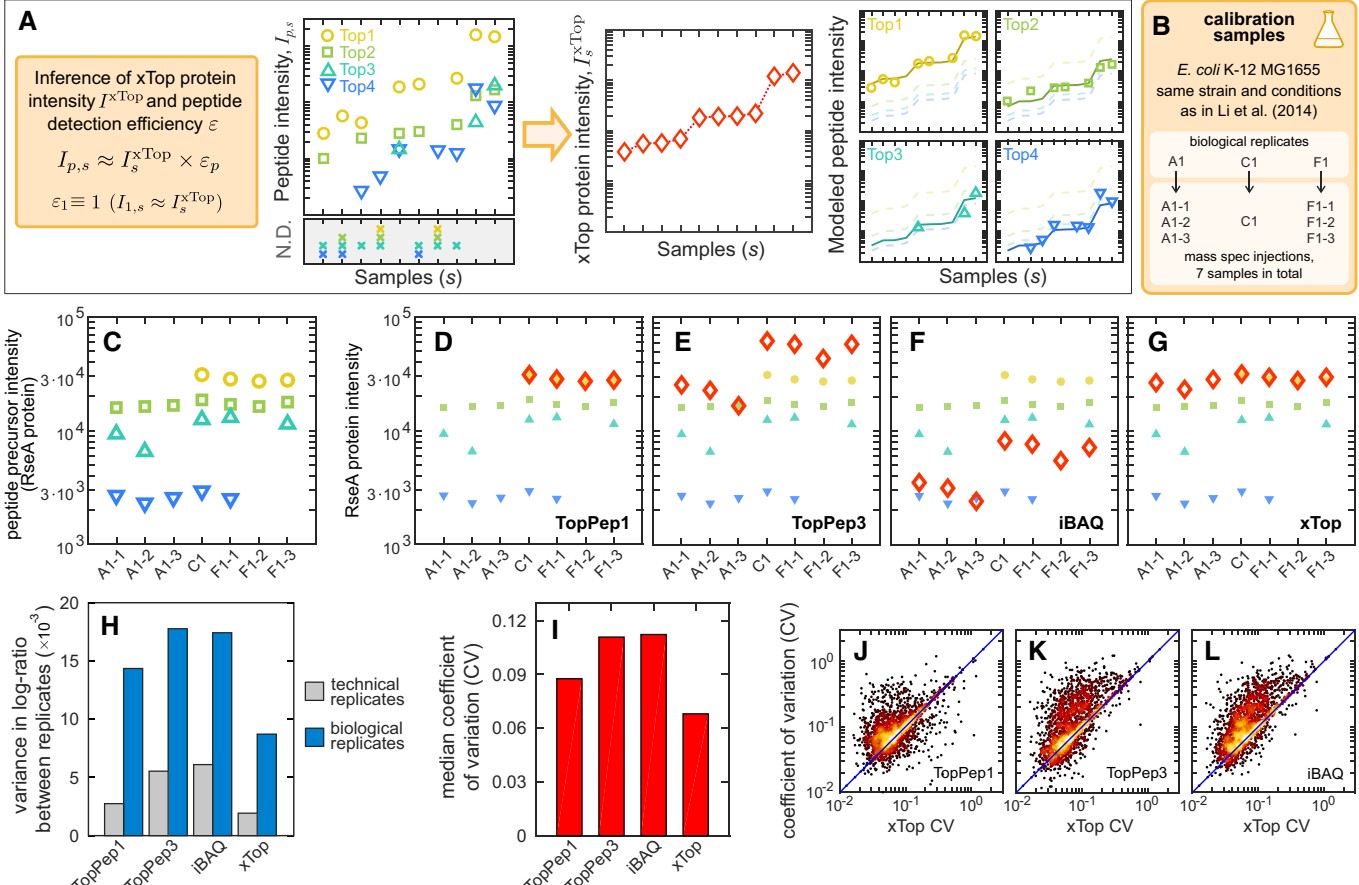

**Figure 2. xTop protein quantification and comparison to other methods.**

A    xTop is a protein inference algorithm which models for each protein the intensities of peptide precursors as the product of the xTop protein intensity $I_s^{\mathrm{xTop}}$ in sample $s$, and a detection efficiency $\epsilon_p$ for each peptide precursor $p$ relative to the peptide with the largest intensity (Top1). This allows to integrate consistently the information from the whole dataset and minimizes the impact of missing peptides on the inferred protein intensity.

B    We collected 3 biological samples of *E. coli* K-12 MG1655 (EQ353) in glucose minimal media, matching strain, and condition from Li *et al* (2014). Two of the three biological replicates were injected 3 times, for a total of 7 proteomics "calibration" datasets. These samples were used for testing the reproducibility of the proteomics measurements and the absolute quantification.

C    Peptide precursor intensities measured for the RseA protein across the seven calibration samples. Different symbols and colors indicate different unique peptide precursors. Peptide-level intensities are reported in Datasets EV4 and EV5.

D–G  Protein intensities (red open diamonds) obtained from the data in panel (C) (also shown in these panels as smaller symbols) computed with four protein inference algorithms: TopPep1, TopPep3, iBAQ, and xTop (Dataset EV6).

H    Variance in the log-ratio of protein intensities between technical (samples F1-1 and F1-2) or biological (samples A1-1 and F1-1) replicates using the same proteins ($N = 1,631$) quantified in all samples by each method (see also Appendix Fig S2).

I    For each protein, the coefficient of variation (CV) of the protein intensities was computed across the seven calibration samples using the same $N = 1,939$ proteins excluding non-detected proteins. The bar graph shows the median CV for each of the four methods employed.

J–L  Scatter of CV computed from TopPep1, TopPep3, and iBAQ against that of xTop. An excess of points is visible above the diagonal (blue line) especially for TopPep3 and iBAQ.

and iBAQ), we grew three replicate cultures of *E. coli* K-12 MG1655 (sub-strain EQ353) cells in minimal medium (MOPS + glucose) in exponential growth (Fig 2B). These "calibration samples" (A1, C1, F1) were measured by DIA/SWATH mass spectrometry using a 64 variable SWATH window acquisition method (Collins *et al*, 2017). For two of the three calibration samples (A1 and F1), we additionally performed three technical MS injection replicate measurements. We analyzed the data from these 7 calibration samples in a peptide-centric way using the *E. coli* spectral library described above and the OpenSWATH software (Rost *et al*, 2014). We obtained quantitative intensity values for 18,731 peptide precursors (Datasets EV4 and EV5). Peptide intensities were strongly correlated between technical and biological replicates, with Pearson coefficients (*r*) above 0.987 and 0.966 for technical and biological replicates, respectively (Appendix Fig S1A). The median coefficient of variation (CV) for technical and biological replicates was 5.5 and 10.8%, respectively (Appendix Fig S1B and C).

To illustrate the effect of missing peptides on the inferred protein intensities, we considered the protein RseA, an anti-sigma factor. As shown in Fig 2C, four peptide precursors (open symbols of different colors) are detected in the seven replicates of the calibration condition. However, not all of them are detected across all 7 samples. In particular, the peptide precursor with the highest intensity (yellow circles) is not detected in the three replicates of sample A1, while peptide precursors represented by the green up-triangle and blue down-triangles are not uniformly detected across the technical replicates. These variabilities strongly impact the protein intensities inferred by the TopPep1/3 and iBAQ algorithms as shown in Fig 2D–F: First, the TopPep1 algorithm only provides a protein intensity in the four samples in which the top peptide has been detected (Fig 2D, red diamonds). This protein is declared as "not detected" in the other three samples. For both TopPep3 and iBAQ (Fig 2E and F, respectively), the missing peptides lead to a considerable scatter in the inferred protein intensities (red diamonds), even though the scatter in the intensities of each of the detected peptides is much smaller. The xTop algorithm combines the intensities of all the detected peptides across these samples; its inferred protein intensities (Fig 2G, red diamonds) show little scatter across all replicates compared to those generated by TopPep1/3 and iBAQ.

Proteome-wide results confirm the expectations from the example above. First, TopPep1 detects on average 1,780 proteins across the calibration samples, about 100 less than the other algorithms (which detect between 1,885 and 1,893 proteins). Both technical and biological replicates are found to be strongly correlated (*r* < 0.98) (Appendix Fig S2A and D). However, a clear improvement of xTop over the other methods is seen when comparing the variance of the ratio of intensities between pairs of replicates, as summarized in Fig 2H (see also Appendix Fig S2B and E). When using either technical and biological replicates, xTop shows the least scatter, while TopPep3 and iBAQ show the most. The improvement of xTop over the other algorithms is also evident when computing the median CV of protein intensities across the seven samples, as shown in Fig 2I, with xTop displaying a CV of 6.8%, about only two-thirds of the CVs compared to TopPep3 and iBAQ (11 and 11.2%, respectively). The additional scatter observed for the TopPep1/3 and iBAQ protein intensities is clearly seen in Fig 2J–L, where we plotted the CV computed from these methods against those computed with xTop. In all cases, an overabundance of points

above the diagonal blue line can be seen, indicating proteins whose intensities are more precisely determined by xTop (i.e., lower CV) than the other methods. As shown in Appendix Fig S2C and F, the increased scatter for TopPep3 seen when comparing pairs of replicates was mostly due to proteins for which some peptide intensity values were missing in one of the samples. iBAQ is similarly affected since it makes use of the sum of all peptide intensities. This issue is overcome by the xTop algorithm thanks to the peptide weighing strategy described above.

### Quantification of absolute protein abundance

Next, we computed absolute protein abundances as fractions of total protein mass in the sample. For that we multiplied the protein intensities for each of the four algorithms tested by the known protein molecular weight and normalized to unity (see Appendix Note S1). These absolute protein mass fractions were evaluated against a set of 29 proteins whose absolute protein abundances had been measured with one representative stable isotope-labeled synthetic peptide per protein (AQUA peptides (Gerber *et al*, 2003); Dataset EV7). The determined absolute protein abundances spanned about 3 orders of magnitude. As shown in Appendix Fig S3A–D, all four protein inference algorithms showed good correlations to the abundances determined using AQUA peptides (*r* > 0.92). On average, we observed a slight 20 to 30% reduction in the median protein abundances that might reflect a discrepancy in the estimation of total protein amount in the sample. When looking at the ratios between the inferred protein mass fractions and the AQUA peptide-derived abundances, we saw that 50% of the data lies in a 2.5-fold range (Fig 3A, red bars), except for iBAQ having the smallest scatter (1.5-fold). Note that 27 of these 29 proteins had been detected with 8 or more peptide precursors, suggesting a minimal impact of fluctuations in the number of detected peptides, which cause most of the scatter in Fig 2.

As our "calibration samples" (A1, C1, F1) were done for the exact same sub-strain (EQ353) of MG1655 and growth condition (MOPS glucose medium) as that analyzed in a previous study (Li *et al*, 2014) by ribosome profiling, we also compared protein abundances against those inferred from Li *et al* (2014) where they determined the protein synthesis rates for more than 96% of all *E. coli* proteins. Since protein degradation is negligible for the vast majority of proteins in exponentially growing *E. coli* cells (Koch & Levy, 1955; Mandelstam, 1958; Pine, 1970; Goldberg & St John, 1976; Erickson *et al*, 2017), synthesis rates are proportional to absolute protein copy numbers. In turn, these can be converted to absolute protein mass fractions (Appendix Note S1). To test whether the synthesis rates obtained via ribosome profiling can be used to obtain absolute protein mass fractions, we compared the mass fractions of the 29 proteins to the AQUA data. We found that ribosome profiling-derived mass fractions correlated with AQUA-measured proteins as strongly as those derived from iBAQ, with 50% of the genes within a 1.5-fold interval (Fig 3A, blue bar and symbols; Appendix Fig S3E). This suggests that ribosome profiling provides good absolute protein quantification.

As an independent test for the accuracy of absolute quantification, we investigated the performance of proteomics- and ribosome profiling-based quantification on proteins expected to participate in protein complexes with known stoichiometric ratios, such as ribosomes, ATP synthase complex (*atp* operon), and NADH

dehydrogenase (*nuo* operon). For each method, we computed the concentration of each protein in the complex (assuming a total protein concentration of $3 \times 10^6$ proteins per $\mu m^3$, see Appendix Note S1) and divided by the stoichiometric coefficient of each protein, obtaining for each protein an estimate of the concentration of the protein complex. If proteins are indeed produced in stoichiometric abundances, the scatter in these ratios should predominantly reflect the error in absolute abundance determination by each method. Fig 3B–E shows the resulting ratios. We observed that ribosome profiling yielded considerably less scatter

compared to the other proteomics methods, in particular for the cases of ATP synthase proteins (Fig 3B) and the ribosomal large subunit (Fig 3E).

Taken together, the results in Fig 3A–E suggest that ribosome profiling provides consistently better absolute quantification than the proteomics methods explored here. Using ribosome profiling as genome-wide standard for evaluating the absolute protein abundances from the various proteomics methods, we found that the > 1,700 proteins measured by DIA/SWATH in the "calibration" samples correspond to about 97% of the total protein mass (as

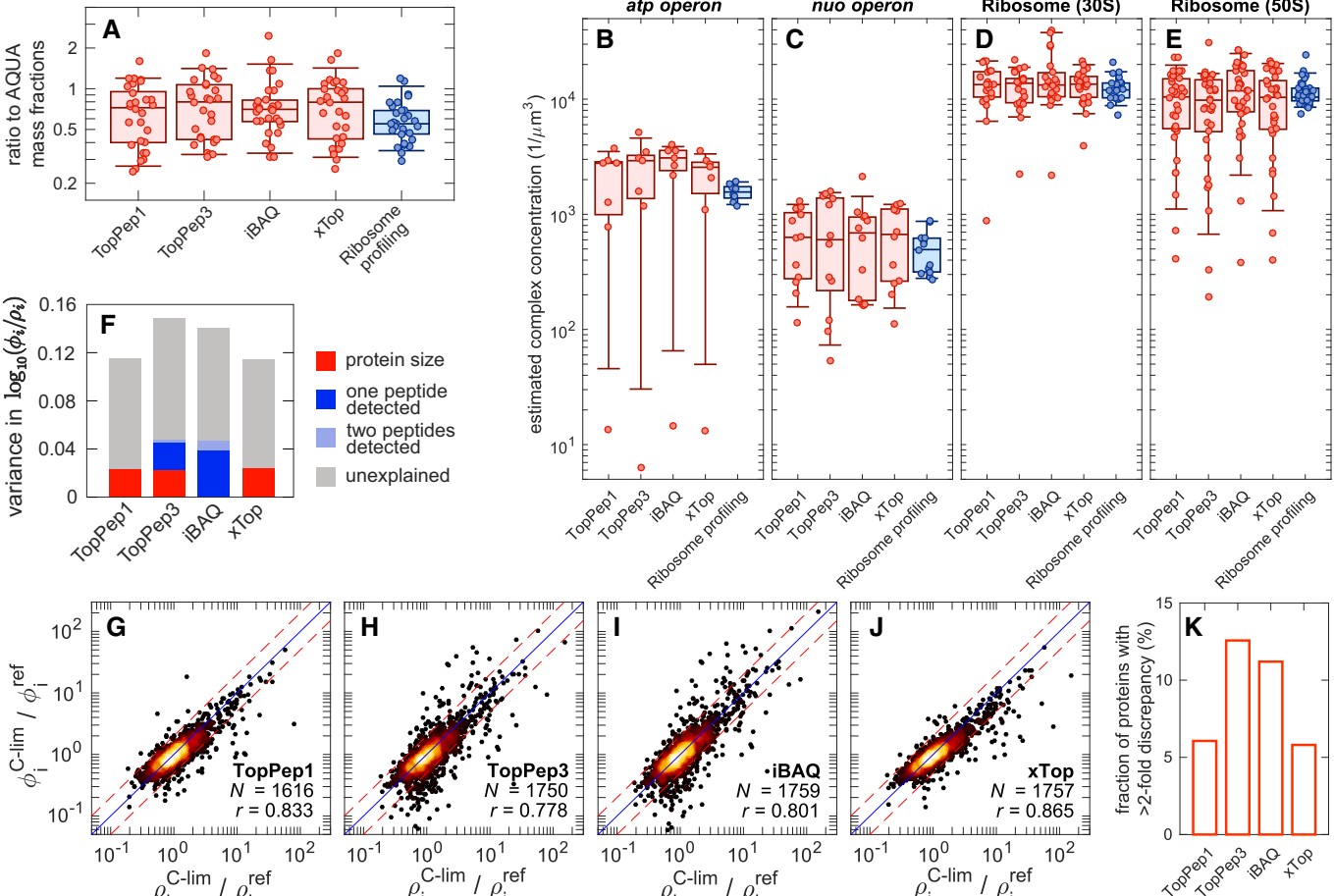

**Figure 3. Relative and absolute quantification.**

A   Ratio of protein mass fractions computed with several protein inference algorithms (x-axis labels) and mass fractions quantified with spiked-in labeled peptides (AQUA) for a set of 29 proteins spanning more than 2 orders of magnitude (see Appendix Fig S3, Dataset EV7). The red boxes and symbols represent four proteomics-based protein inference algorithms (TopPep1/3, iBAQ, and xTop). The blue box and symbols correspond to ribosome profiling-derived mass fractions. The boxes and whiskers include 50 and 90% of the data, respectively, while the central line represents the median.

B–E  Estimated concentration of protein complexes from various protein inference algorithms and ribosome profiling. Individual points are the estimated concentrations for individual proteins in the complex, divided by the number of copies in each complex. Individual concentrations were estimated from the observed protein number fractions assuming a total protein concentration of $3 \times 10^6$ proteins/$\mu m^3$ (Appendix Note S1). The boxes and whiskers include 50 and 90% of the data, respectively, while the central line represents the median.

F   The bar chart summarizes the multilinear analysis described in Appendix Note S3. The total height of the bar is the variance in the log-ratio of proteomics-based and ribosome profiling-based mass fractions. Colored components represent the fraction of variance explained by each factor (red, protein size; blue, only one peptide precursor detected; light blue, only 2 peptide precursors detected). TopPep1/3 and xTop protein mass fractions display a bias toward large proteins; TopPep3 and iBAQ systematically underestimate the abundance of proteins with 1 or 2 detected peptides.

G–J  Scatter plot of fold change between reference and carbon-limited conditions (growth rates 0.91/h and 0.35/h, respectively) in mass spectrometry-based ($\phi_i^{C-lim}/\phi_i^{ref}$, y-axis) and ribosome profiling-based protein mass fractions ($\rho_i^{C-lim}/\rho_i^{ref}$, x-axis). The blue line represents equal changes in the two quantities; the dashed lines represent a 2-fold discrepancy.

K   Fraction of proteins showing a > 2-fold discrepancy in panels (G–J).

estimated by data from ribosome profiling) and captured most proteins with mass fractions above $10^{-5}$ (Appendix Fig S4A). However, comparisons with ribosome profiling also revealed systematic biases in the absolute quantification for each of the protein inference methods, which were quantified by a multilinear regression of the log-ratio of proteomics and ribosome profiling-derived protein mass fractions, denoted as $\phi_i$ and $\rho_i$ respectively, as described in Appendix Note S3. The amount of bias (explained variance) is summarized in Fig 3F: TopPep3 and iBAQ are found to underestimate the abundances of proteins detected with only one or two peptide precursors (blue and light blue in Fig 3F). As the number of detected peptide precursors is typically very low for proteins with mass fraction below $10^{-4}$ (Appendix Fig S4B and C), this bias leads TopPep3 and iBAQ to underestimate the abundance of low-abundant proteins compared to ribosome profiling or xTop/TopPep1 (Appendix Figs S4D–E and S5). On the other hand, TopPep1/3 and xTop are seen to overestimate the absolute abundances of larger proteins, leading to systematic difference of 2- to 3-fold between large and small proteins (red color in Fig 3F; Appendix Fig S4E–G). As a consequence, proteomics algorithms tend to overestimate the covered protein concentration range in a given organism (difference between the measured concentrations of the highest and lowest abundant protein).

### Quantification of relative protein abundance

The systematic biases in absolute quantification displayed above do not necessarily impact relative quantification. In particular, protein-specific biases such as that toward protein size can potentially be "fixed" by a condition-independent scaling factors (Appendix Note S3). To assess the relative quantification capabilities of xTop, TopPep1, TopPep3, and iBAQ, we performed additional ribosome profiling measurements in both glucose minimal medium and carbon-limited growth for *E. coli* NCM3722 strain, for which we also have performed proteomics analysis (Dataset EV10). We then compared the fold change in protein mass fractions estimated from proteomics ($\phi_i^{C-\lim}/\phi_i^{\mathrm{ref}}$) to that estimated from ribosome profiling ($\rho_i^{C-\lim}/\rho_i^{\mathrm{ref}}$), as shown in Fig 3G–J. Proteins lying on the diagonal (blue line) had matching fold changes in proteomics and ribosome profiling mass fractions. Red dashed lines indicate 2-fold differences between the two. Among the four proteomics methods investigated, xTop showed the strongest correlation with ribosome profiling ($r = 0.865$). TopPep1 displayed a slightly lower correlation than xTop ($r = 0.833$), and 10% less quantified proteins (1,616 for TopPep1 vs 1,757 for xTop). TopPep3 and iBAQ had the lowest correlation coefficients ($r = 0.778$ and $0.801$, respectively). The degree of scatter is quantified in Fig 3K, which represents the fraction of proteins for which the fold change in proteomics $\phi_i^{C-\lim}/\phi_i^{\mathrm{ref}}$ differs from that of ribosome profiling $\rho_i^{C-\lim}/\rho_i^{\mathrm{ref}}$ by more than 2-fold (red dashed lines in Fig 3G–J). For both TopPep3 and iBAQ, the fraction is 12 and 11%, respectively, twice of that obtained for the other two methods (~ 6%). The additional scatter is likely caused by the missing peptides (Fig 2E, Fig N2.1 in Appendix Note S2) and the nonlinear dependence on protein abundance (Appendix Figs S4D–E and S5), which both impact relative quantification.

### A versatile workflow for absolute abundance quantification across many conditions

The above analysis suggested that the xTop algorithm is superior to TopPep1/3 and iBAQ algorithms in inferring protein intensities and

in capturing the relative protein abundance across conditions. However, ribosome profiling is superior to the proteomics-based methods in quantifying absolute protein abundances in *E. coli*, i.e., an organism for which protein degradation can be neglected under exponential growth. Given the costs and efforts of performing ribosome profiling across many conditions, we developed a workflow combining the accuracy of ribosome profiling with the versatility of mass spectrometry. As illustrated in Appendix Fig S6, we used DIA/ SWATH-based proteomics to measure all our *E. coli* samples and xTop to obtain protein abundances relative to the 7 calibration samples, for which ribosome profiling data are available (Li *et al*, 2014). We then used this ribosome profiling data to rescale the relative proteomic data to absolute protein mass fractions. Note that the use of a calibration sample allows different proteomics datasets to be combined in a consistent way, as long as each dataset includes the same "calibration" condition. In this workflow, there might be additional sources of error, for example, for membrane-associated or periplasmic proteins. These proteins can be problematic since they translocated across membranes and protein extraction efficiencies might not be 100% and vary across conditions. However, a comparison of the absolute protein mass fractions determined by ribosome profiling and xTop showed that the ratio was centered around 1 for different classes of membrane and periplasmic proteins (Appendix Fig S7). This indicates that at least in the conditions tested extraction efficiencies of membrane-associated proteins are not an issue.

Our generated data, including the comprehensive *E. coli* spectral library and the absolute protein mass fractions obtained through xTop, provide accurate quantitative estimates for thousands of individual proteins across > 60 growth conditions, a task which is not practical to achieve by ribosome profiling. For other conditions and organisms (e.g., slow-growing bacteria, or eukaryotic cells), the impact of protein degradation and protein translocation might not allow the use of ribosome profiling to calibrate relative protein abundances. In these cases, our versatile workflow can be adapted to use different quantities for the calibration of absolute protein abundances (see Appendix Note S3, section "Bias removal via calibration of protein abundances").

### Biological analysis of *E. coli* proteomes

The workflow outlined above allowed us to analyze *E. coli* proteomes over 66 different samples representing an array of different treatments, strains, and growth conditions. The resulting absolute protein abundances are expressed in "protein mass fractions", i.e., mass of a given protein over the total mass of all detected proteins, which can readily be converted to cellular protein concentration (Appendix Note S1). Assuming typical proteins to be 300-residues in length, 0.1% of proteome mass is equivalent to about 2,400 proteins per $\mu m^3$ of cellular volume in *E. coli*. Note that the frequently used absolute unit "protein copies/cell" is avoided here, as cell size is highly variable across growth conditions (Basan *et al*, 2015b; Si *et al*, 2017). Instead, protein mass fractions allow direct conversion to protein concentrations, which can then be compared across conditions (Milo, 2013).

Altogether, we obtained absolute mass fractions for a total of 2,335 distinct proteins across all samples, covering the vast majority of the expressed *E. coli* proteome. The 66 samples analyzed can be

divided into three groups. First, seven samples correspond to the "calibration" samples discussed above. Second, 27 samples define three "growth limitation" series, in which cells were growing exponentially in glucose minimal medium, with various means of titration to implement gradual growth limitation in catabolism, anabolism, and protein synthesis as done previously (Hui *et al*, 2015). This dataset is used to define "protein sectors", which provide a coarse-grained view of changes in the *E. coli* proteome. Third, 30 samples are analyzed to reveal the proteome response to various stresses, during transition from exponential growth to stationary phase, and away from planktonic state, e.g., for growing colonies in biofilm-forming conditions. All sample descriptions are listed in Datasets EV2 and EV3, while the absolute protein mass fractions are reported in Datasets EV8 and EV9.

## Adaptation of coarse-grained protein sectors

Following Hui *et al* (2015), we applied a variety of steady-state growth limitations to *E. coli* K-12 NCM3722 and its derivatives. First, culturing cells with titratable glucose uptake (strains NQ1243 (Basan *et al*, 2015a) and NQ1390; see Dataset EV1 and Extended Experimental Methods for strain informations) in glucose minimal medium, we generated a series of 15 samples that grew at a range of growth rates (0.33–0.91/h); they are referred to as the "C-limitation" series. Second, culturing cells with titratable glutamate synthesis (NQ393, Hui *et al* (2015)) in the same medium (with ammonium as the sole N-source), we generated a series of seven samples that grew from 0.22–0.84/h due primarily to the effect of glutamate on amino acid synthesis through trans-amination (Reitzer, 2005). They are referred to as the "A-limitation" series. Finally, by adding various sub-lethal doses of chloramphenicol into the growth medium (again, glucose minimal medium), we generated another seven samples that grew from 0.36–0.98/h due to limitation of protein synthesis by the ribosomes; they are referred to as the "R-limitation" series.

Changes in protein abundances in response to these applied growth limitations were characterized previously by Hui *et al* (2015) using a simple binary classification, which partitions the proteome into a number of "sectors". For instance, proteins upregulated in C-limitation and downregulated in A- and R-limitation define the "C-sector". The resulting $2^3 = 8$ possible protein sectors are summarized in Fig 4A, which also shows the number of genes belonging to each sector. Our data allowed us to associate 1,821 of the 2,335 proteins detected to the eight protein sectors, using only proteins with at least three data points in each growth limitation series (Dataset EV11). A high degree of consistency was seen in the classification of the proteins classified here and in Hui *et al* (2015) (Appendix Fig S8). Most discrepancies were due to proteins weakly dependent on growth rate in at least one of the three growth limitations, which were therefore borderline between pairs of sectors (Appendix Fig S8H). The total number of proteins classified here was almost double from that reported in Hui *et al* (2015), although the latter represent close to 90% of the total protein mass detected in this work, indicating that most of the newly classified proteins were low in abundance. The total abundance of proteins belonging to each sector, i.e., the sector mass fraction, is given in Fig 4B for the reference condition (glucose minimal medium).

At a quantitative level, the C-, A-, S-sectors, which are upregulated in response to C- and/or A-limitation, are at about 10% of

proteome mass each in reference condition (glucose minimal medium) and reach either half or twice the proteome fraction at the slowest growth rate examined for each limitation (Fig 4C–E). The U-sector, which is downregulated in all three limitations, ranges between 20% of the proteome in reference condition to less than 10% at slow growth (Fig 4F). The R-sector, which is the largest sector in reference condition at 30% of the proteome, is upregulated under R-limitation reaching about 45% of the proteome (Fig 4G). These overall patterns of these sector abundances resemble well those observed in Hui *et al* (2015).

Following Hui *et al* (2015), we applied a GO-term enrichment analysis to proteins in each sector to bring forth common functional roles (Appendix Note S4). The analysis in Hui *et al* (2015) focused on only five of the eight proteins sectors, as these displayed the largest variation in expression across the different growth conditions: C-sector ($C^\uparrow A^\downarrow R^\downarrow$, predominantly associated with carbon catabolism and motility), A-sector ($C^\downarrow A^\downarrow R^\downarrow$, glycolysis and amino acid biosynthesis), R-sector ($C^\downarrow A^\uparrow R^\uparrow$, ribosomal, and ribosome-associated proteins), S-sector ($C^\uparrow A^\uparrow R^\downarrow$, catabolism and stress response), and U-sector ($C^\downarrow A^\downarrow R^\downarrow$, biosynthesis of amino acids and nucleotides). Our findings were in excellent agreement with the previous analysis, with similar sector membership (Appendix Fig S8) and functional roles for each of these five sectors (Dataset EV12).

Proteins not belonging to any of the five sectors above (171 out of 1,034 in Hui *et al* (2015)) had been previously lumped into a single group (the "O-sector"). In our study, a much larger number of proteins (686 out of 1,821 classified), including most of the newly detected proteins, belonged to this group. While these proteins comprised a total of 20–30% of the proteome mass (Fig 4H, 12–20% for the O-sector proteins in Hui *et al* (2015)), most of them had low individual abundances, with 95% of them below 0.01% of the total proteome mass (Appendix Fig S9). This group of proteins could be further categorized according to their responses to the three growth limitations into the C'-sector ($C^\uparrow A^\downarrow R^\uparrow$, 103 genes), A'-sector ($C^\downarrow A^\downarrow R^\uparrow$, 211 genes), and S'-sector ($C^\uparrow A^\uparrow R^\uparrow$, 372 genes, see Fig 4A and B). Each of these sectors included proteins which were upregulated under R-limitation and additionally C- and/or A-limitation. Their abundances across growth conditions, shown in Fig 4I–K, were dominated by the S'-sector, which rose from 10% of the proteome in reference condition to as much as 20% at slow growth. More than half of this increase was accounted for by changes in two abundant proteins, Lpp and OmpC (Dataset EV9).

The GO-term enrichment analysis (Fig 4L; Appendix Note S4; Dataset EV12) found several biological activities to be shared among these three sectors and some of the other five sectors described above. In particular, the C'-sector included many amino acid, peptide, and protein transporters, which were upregulated in C-limitation and downregulated in A-limitation. This, together with their generally weak response to R-limitation, made the C'-sector proteins very similar to the C-sector in both their response and function (higher-order functional groupings are indicated by the dashed lines in Fig 4A). Both the A' and the S' sectors were enriched in proteins involved in "cell division" and "cell cycle", as well as in a variety of terms associated with cell membrane and cell wall (Fig 4 L). Additionally, the A'-sector was enriched in "tRNA processing", "rRNA processing" terms, due to proteins involved in ribosome biogenesis (*rlm*, *rsm* operons, whose proteins where hardly detected

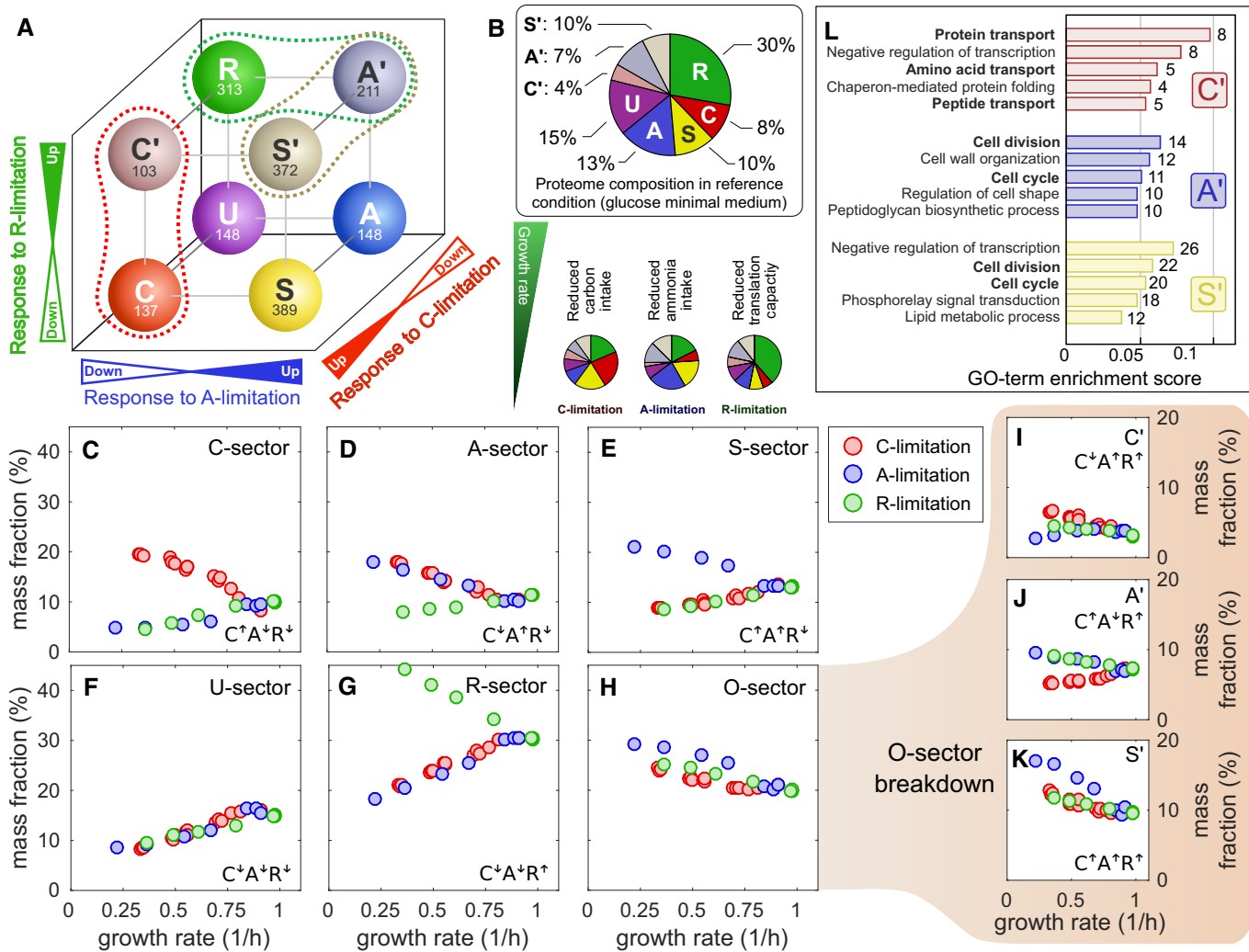

**Figure 4. Proteome sectors in carbon, nitrogen, and translational limitation.**

A   Starting from a "reference" condition (glucose minimal media, growth rate approx. 1/h), we modulate *E. coli* growth by applying three different sources of stress ("limitations"): carbon (C-) limitation, obtained by titrating glucose transport; anabolic (A-) limitation, obtained by titrating nitrogen assimilation; ribosomal (R-) limitation, obtained by introducing translation-inhibiting antibiotics (chloramphenicol). The simplest way to capture the change in the proteome composition is to introduce a binary classification: A protein can be either up- or downregulated in each of the three limitations. For example, the C-sector includes all proteins whose abundance increases in C-limitation and decrease in A- and R-limitations, and is hence indicated as C↑A↓R↓. In the diagram, we show the eight possible sectors, with the number of genes associated with each sector. Dashed lines indicate higher-order groupings between sectors with partially overlapping GO-terms.

B   The decomposition of the proteome into protein sectors allows to appreciate the large-scale changes of protein expression in the three growth limitations. The pie charts indicate the composition of the proteome by mass in reference condition and in the three extreme limitations (growth rate approx. 0.3/h).

C–K   Protein mass fractions associated with each of the eight sectors across growth rates in the three growth limitation series. Panel (H) represents the mass fraction of the O-sector, given by the sum of C'-, A'-, and S'-sector protein abundances (panels (I) to (K)).

L   GO-term enrichment analysis of the C'-, A'-, and S'-sectors (see Appendix Note S4). The numbers at the side of the bars represent the number of genes in the sector associated with each GO-term. Bold terms are mostly unique to each one of the three sectors, while the others are shared between two of the three sectors.

in Hui *et al* (2015)), and "response to antibiotic", causing it to also functionally overlap with the R-sector.

The comprehensive coverage in our dataset allowed us to investigate the expression profile of proteins belonging to each of the 8 sectors. Surprisingly, we found that different sectors have markedly different protein abundance distributions (Appendix Fig S9A–I). The composition of both the S- and S'-sectors was strongly skewed toward low-abundant proteins, with 62% of genes having a mass fraction $\phi_i < 10^{-4}$ in reference condition, while most genes

belonging to the A- and U-sectors were relatively abundant (only 32% of genes below $10^{-4}$) (Appendix Fig S9J). Overall, low and high abundant proteins seemed to respond to nutrient starvation in qualitatively different ways: About half of the low-abundant proteins (42% of genes with mass fractions $\phi_i < 10^{-4}$ in reference condition) were upregulated under both nutrient limitations (either carbon or nitrogen), while 52% of highly expressed genes ($\phi_i > 10^{-3}$), including the majority of biosynthetic enzymes, tended to be downregulated in both nutrient limitations (Appendix Fig S9K).

In each of the C'-, A'-, and S'-sectors, one abundant protein stood out against the above pattern. This were the outer membrane porins: OmpF (C'-sector), OmpA (A'-sector), and OmpC (S'-sector), which together with NmpC (C-sector) responded to the three growth limitations with very different logic (Fig 5A–C). The variety of expression patterns exhibited by the different porins is shown in Fig 5A–C. In Fig 5D, we compared to their abundances in high osmolarity medium, which is known to induce big shifts in porin expression (Alphen & Lugtenberg, 1977). OmpA is the basal porin, expressed at high levels in all conditions except for C-limitation. OmpF is expressed in all conditions except for high osmolarity. Interestingly, OmpC and NmpC seem to complement each other, with NmpC specializing in C-limitation and OmpC specializing in high osmolarity. The high expression of NmpC in poor carbon sources was overlooked in previous studies due to the loss of *nmpC*

in certain *E. coli* K-12 strains (Hindahl *et al*, 1984), as will be discussed below.

The total mass fraction of these porins is shown in Fig 5E, with an increase under C- and A-limitation and a decrease under R-limitation. This change is echoed in the mass fraction of periplasmic proteins, although with twice as large increase under C-limitation (Fig 5F). Although the increasing trend of periplasmic protein expression in carbon limitation was already noted in Schmidt *et al* (2016), they reported a much larger fraction of periplasmic proteins, approaching 1/3 of total cytoplasmic proteins at slow growth, and giving a rather different biological picture.

The increase of porin protein abundance under C-limitation coincides with the increase of cell surface area due to reduced cell size observed in poor nutrient conditions (Basan *et al*, 2015b; Si *et al*, 2017). To see whether the data in Fig 5E can be explained by

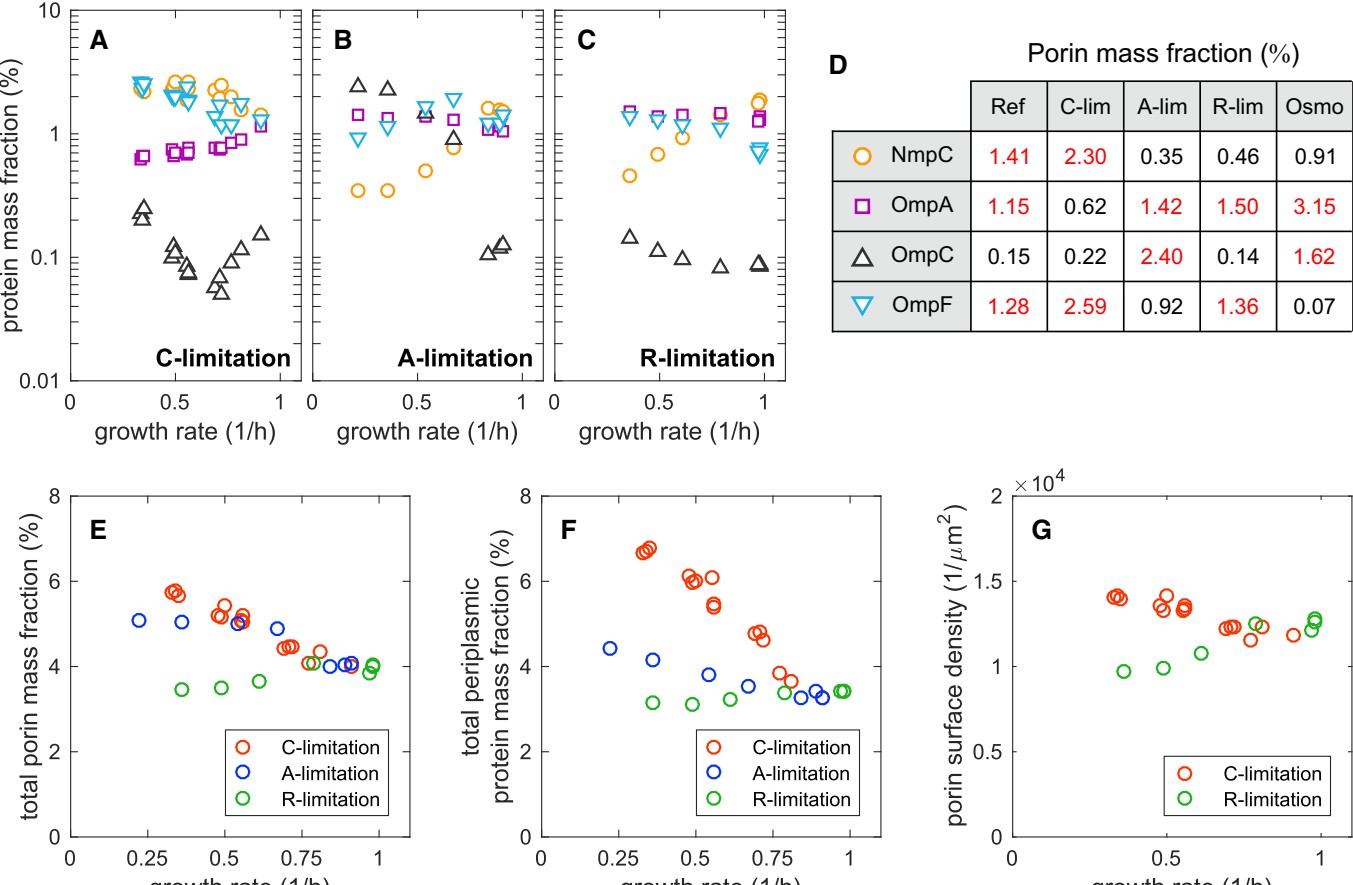

**Figure 5. Outer membrane proteins.**

A–C   Absolute protein mass fractions (in % of total protein mass) of the four most abundant outer membrane porins in *E. coli* NCM3722 (NpmC, OmpA, OmpC, and OmpF), as a function of growth rate in three growth limitation series; symbols are shown in the adjacent panel.

D   The table summarizes the mass fractions of the four porins in reference condition (glucose minimal medium), extreme C-, A-, and R-lim (growth rate ~ 0.3/h), and high osmolarity (sample Lib-02, growth rate 0.24/h). Entries with mass fractions larger than 1% are highlighted in red.

E   Total protein mass fractions of the four porins against growth rate, in the three growth limitation series.

F   Total protein mass fraction of proteins classified as "periplasmic" according to Ecocyc classification (Karp *et al*, 2018) against growth rate, in the three growth limitations.

G   Porin surface density in C-limitation (red) and R-limitation. To obtain the porin surface density, we first obtained the number of porins per cell volume (see Appendix Note S1) and divided by the (condition-dependent) surface to volume ratio, using the values reported by Si *et al* (2017) for C- and R-limitation.

   

changes in surface-volume ratio, we computed the total density of porins on the outer membrane using the cell size from Basan *et al* (2015b) for C- and R-limitation and assuming a constant aspect ratio (Si *et al,* 2017) (Fig 5G). In reference condition, we obtained a density of about $1.2 \times 10^4$ porins/$\mu m^2$, corresponding an occupation of 5% of the cell surface area given a diameter of 2.6 nm for each porin (based on the structure of the OmpA monomer (Pautsch & Schulz, 1998)). The density increased by 15% under carbon limitation and decreased by 20% under chloramphenicol treatment. These results demonstrate that the total porin density is not conserved. Rather, the porin density on the surface is low enough that it can be adjusted in accordance to physiological demand.

## Proteome response under diverse growth conditions

In addition to the global, sector-level analysis performed so far, the accuracy of absolute protein mass fractions attained by our workflow allowed us to make quantitative statements on the absolute abundances of *individual* proteins across conditions. In this section, we turned our attention to such an effort. We divided the analysis into four categories. First, we compared the differences in proteome response between growth on different carbon sources and on glucose with titratable uptake. Second, we analyzed proteome response under various nutrient-limiting conditions beyond carbon and nitrogen, including exponential growth under phosphate limitation and anaerobic condition, as well as transition to stationary phase, colony growth, and biofilm-inducing conditions. Third, we examined response to various stress conditions under nutrient replete conditions, e.g., under high temperature, high osmolarity, and oxidative stress. While the above are all done with NCM3722 cells and their close derivatives, we finally compared the proteome of these with a number of other genotypes, which gave us the opportunity to compare expression of motility and biofilm-associated genes, which are often subject to mutations in laboratory strains of *E. coli*.

### Different carbon sources

In this study, we measured the proteome of cells grown in a variety of carbon substrates (including mannitol, melibiose, arabinose, xylose, gluconate) that are not commonly used for *E. coli* proteomic studies. As these substrates provided a range of growth rates, this dataset presents a unique opportunity to examine the effect of "carbon limitation" imposed by these individual substrates, compared to "C-limitation" described in previous sections imposed by titrating the glucose transporter PtsG for cells grown on glucose. This is done in Fig 6A by showing the increase in the abundance of proteins for cells grown in each of these substrates, versus glucose-limited growth dialed to obtain similar growth rate. Shown in red are proteins that increased by at least 4-fold in relative abundance and by at least 0.05% of the proteome in absolute abundance (~ 500 copies out of a million assuming proteins are of similar length). The first thing to notice is how few such "red proteins" there are out of ~ 2,000 total proteins detected. Among these red proteins, about half are specific to the catabolism of the particular substrate being used, e.g., MtlAD for mannitol and MelA for melibiose. It is striking that none of the specific catabolic proteins increase in expression by more than a few percent of the proteome. The highest expressed proteins, MelA (α-galactosidase) and LacZ (β-galactosidase), are at 4

and 1.3% of the proteome, respectively, when grown on melibiose, an α-galactoside. The others increased by not more than 1% of proteome. Additional catabolic proteins associated specifically with the catabolism of the particular substrates are indicated in blue. Their abundances are mostly lower than the red proteins; the few exceptions with increase > 0.05%-proteome have the gene name indicated in blue, including UhpT (G6P), AraAB (arabinose), XylB (xylose), and GlpDTQ (glycerol). (They are not shown in red because the relative abundance changed less than 4-fold, indicating appreciable expression also under glucose-limiting growth.)

The sum of the abundances of all the catabolic proteins specific to each respective carbon sources, i.e., all the proteins indicated in blue, are plotted as filled diamonds in Fig 6B and comprise only 1–2% of the total proteome by mass. [The one exception is for growth on melibiose, green diamond, with MelA alone being 4% of the proteome. However, the abundance of MelA was assigned without the benefit of calibration by ribosome profiling (as these proteins were not detected in the calibration sample grown in glucose medium) and may therefore be less accurate.]. The small increase in these carbon source-specific catabolic proteins is to be contrasted with the substantial increase of the total abundance of "C-sector proteins", as defined from the three growth limitation series described above, plotted as filled circles for cells grown on the specific carbon sources and as red open circles for cells grown on glucose with titrated. These two datasets are remarkably similar, indicating that the bulk of the C-sector proteins respond primarily to the rate of carbon-limited growth, not to the specific carbon substrates. Quantitatively, those responding to specific carbon sources, filled diamonds, represent only 5–10% of C-sector proteins by mass. The similarity between glucose limitation and the variety of other carbon sources actually extends beyond C-sector proteins, as the abundances of the other protein sectors are also similar; see Appendix Fig S11 for an overview of the abundances of protein sectors across all conditions studied in this work. Together, these results establish the concept of "carbon-limited growth" and validate the use of the titratable glucose uptake as a method to probe the proteome response to carbon-limited growth.

Among the C-sector proteins, the abundances of TCA/gluconeogenesis (GNG) enzymes and flagellar components each comprise about 25% of the total sector abundance (orange and blue symbols, Fig 6C). While for most of these carbon substrates studied, the abundance of these TCA/GNG and flagellar proteins are similar to those observed for the titratable glucose uptake strain (Fig 6D and E; compare red and gray symbols), there are several exceptions worth noting. Cells grown on acetate (blue circles) exhibit a significant increase (+ 3% proteome) in the abundance of C-sector TCA/GNG enzymes; this is expected as these enzymes play a key role in the assimilation of acetate. Interestingly, this is accompanied by a decrease in the abundance of flagellar proteins, also about 3%. We also note that for growth on melibiose, flagella proteins are not expressed (Fig 6E, green circle), this is due to the use of strain EQ59, a derivative of NCM3722 abolished of flagella expression as will be discussed below. (For all other cases of growth on alternative carbon substrates, NCM3722 was used). The lack of flagella expression leads to ~ 3.5% reduction in the total abundance of C-sector proteins (Fig 6B, green circle). Recalling that these C-sector proteins are defined from titratable glucose uptake and do not include the melibiose-specific enzymes, particularly MelA and LacZ,

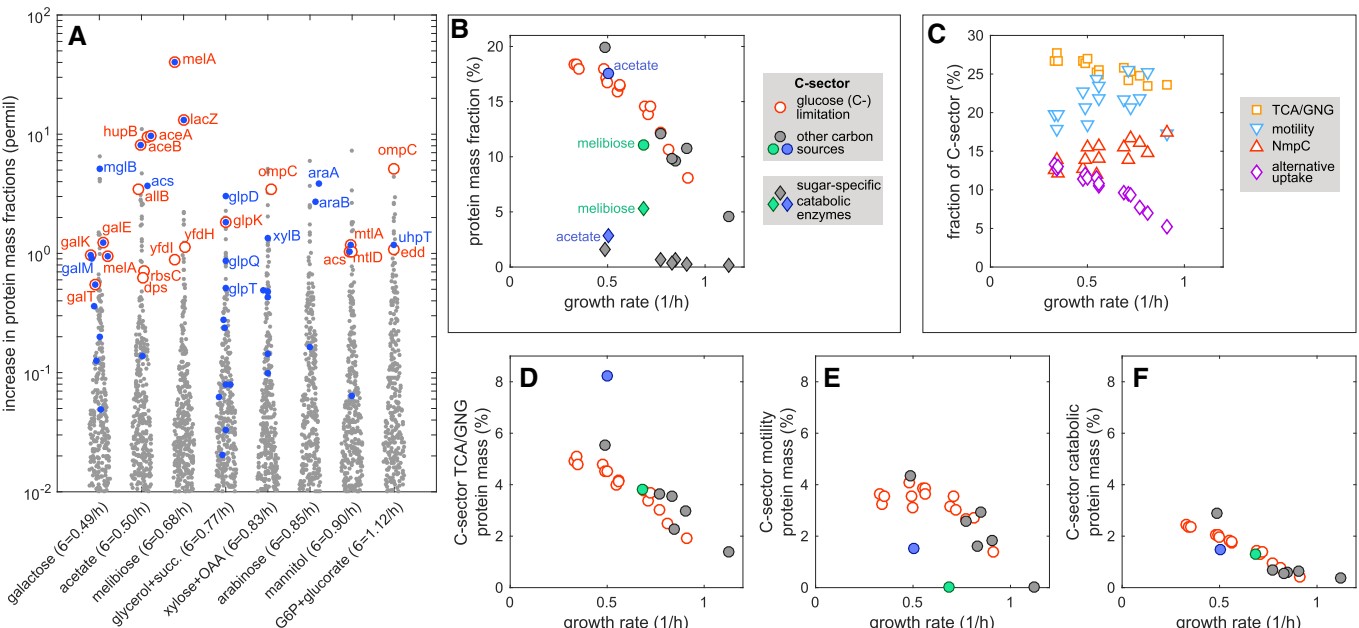

**Figure 6. Carbon-limited growth.**

A   Comparison between the proteome of *Escherichia coli* cells grown in various carbon sources and that of cells grown in glucose. For each carbon source (x-axis; the growth rate is specified in the label), we show the increase in protein mass fractions in that medium from the abundances observed for cells growing at a similar rate under "C-limitation", i.e., on glucose with titratable expression of the glucose transporter PtsG in NCM3722-derived strains NQ1243 and NQ1390; see Dataset EV1. For growth on carbon sources different from glucose, wild-type NCM3722 cells were used in all cases, except in the case of melibiose where the non-motile NCM3722-derived strain EQ59 was used instead. Red circles indicate proteins with a fold increase larger than 4; blue dots represent proteins associated with catabolism of the carbon sources in which cells are grown in (e.g., *glp* genes for growth on glycerol); gene names are displayed for proteins above a mass fraction of 0.5. Proteins that were systematically different across all samples were excluded from the comparison; this was done by computing the differences of the log-transformed mass fractions across all pairs of samples, and excluding proteins for which the absolute value of the mean difference is larger than its standard deviation.

B   Circles indicate the total mass fraction of proteins assigned to the C-sector for cells growing under glucose uptake titration (C-limitation, open red) and for cells grown on the carbon sources described in panel (A) (filled gray, except for growth on acetate, blue, and melibiose, green). Filled diamonds indicate the total mass fraction of proteins associated with the catabolism of specific carbon sources used (those indicated by the blue dots in panel (A)). They amount to only 5–10% of C-sector proteins, with the exception of those associated with the catabolism of melibiose, with a single protein MelA comprising of 4% of total protein mass, or ~ 1/3 of C-sector proteins.

C   Open symbols indicate the fractional abundance within the C-sector of four protein groups (TCA cycle and gluconeogenesis (TCA/GNG); motility; the outer membrane porin NmpC; alternative carbon or amino acid uptake and catabolic proteins) for cells growing under glucose uptake titration. The total protein mass fraction of these four groups adds up to between 64 and 75% of the whole C-sector protein mass.

D–F   Absolute protein mass fraction of TCA/GNG, motility, or alternate carbon uptake proteins that belong to the C-sector. Symbols are the same as in panel (B). Cells growing on acetate (blue) display higher TCA/GNG and lower motility expression levels compared to glucose-limited growth at the same growth rate. Expression of motility proteins is instead greatly reduced for cell growing on melibiose (green), due to the fact that the non-motile strain EQ59 was used in this case.

adding back these two enzymes again makes the total comparable to that of the glucose titration strain. These data suggest a possible compensatory regulation among C-sector proteins under carbon-limited growth.

The remaining C-sector proteins are comprised largely of the outer membrane porin NmpC and a number of catabolic proteins not related to the provided carbon substrates (open red circles and filled purple diamonds in Fig 6C, respectively). As discussed above, NmpC is the major outer membrane porin in C-limitation along with OmpF. Catabolic proteins detected at appreciable levels include those involved in the uptake of amino acids (*cstA*, *dppF*, *livFGM*, *gltJK*, *putP*, *tcyJ*, *yifK*), galactose (*galEKT*, *mglABC*), ribose (*rbsABCDK*), and acetate (*acs*). The abundance of these proteins increase sharply under both glucose uptake titration and the alternative carbon substrates (Fig 6F), suggesting a general foraging strategy in poor carbon conditions. Together, the data in Fig 6 provide a

detailed quantitative picture of the proteome response to carbon limitation, with most resources devoted to general foraging (including flagella, TCA/GNG, and the assortment of catabolic proteins), and with only a small fraction associated with the uptake of the specific carbon substrate provided.

### Other nutrient conditions

In the first part of the results, we applied titratable limitations to probe the response to perturbations of major proteomics sectors (Figs 4 and 5). Here, we explored the proteome response to other nutrient conditions, including anaerobic growth, phosphate limitation, and slowdown into stationary phase. To display the finer differences due to these perturbations, we used scatter plots showing the absolute abundances of all proteins detected in a pair of growth conditions, highlighting a number of major biological functions with distinct colors and symbols. Examples are shown in Appendix Fig

S10 for each of the three limitation series described in the previous section, compared to the reference condition (wild type in glucose minimal medium). Significant movement away from the diagonal was seen for ribosomal proteins (green points), TCA enzymes (open orange squares), glycolytic enzymes (open blue squares), fermentation enzymes (open purple diamonds), motility proteins (cyan triangles), and a subset of stationary, RpoS-driven, proteins (yellow triangles). The overall change in proteome composition in all three cases was similar, close to 25% (Appendix Fig S11).

### MOPS versus M9

For the comparison in Fig 6, samples with specific carbon sources were mostly grown in MOPS-based media, while the titratable glucose uptake strain was grown in M9 glucose medium. Here, we compare NCM3722 grown in MOPS- and M9-based medium, both with glucose as the sole carbon source. The resulting scatter plots are shown in Fig 7A. The vast majority of genes were very similar in expression across conditions, as also clear when looking at

aggregate measures of proteome similarity (Appendix Fig S11). However, there were several notable differences between the two growth media, which we highlighted by the red symbols. First, the *thiCEFSGH* operon, encoding enzymes of the thiamine biosynthesis pathway, was strongly repressed in M9 (undetected). This reflects the supplementation of thiamine in the M9 medium used. Second, the *ent* genes, encoding biosynthesis of enterobactin employed in iron uptake, were significantly upregulated in M9 (2 to 4-fold), while proteins for iron storage (encoded by *ftnA*, *bfr*) as well as the iron-based superoxide dismutase (encoded by *sodB*) were significantly upregulated in MOPS (> 2.5-fold). The iron content in M9 and MOPS is actually not so different (10 µM of $FeCl_3$ in M9, 60 µM of $FeSO_4$ in MOPS). The opposite proteome response observed here is likely due to the addition of 4 mM tricine in MOPS, which solubilizes iron (Neidhardt *et al*, 1974). Consistent with this, small precipitates were observed when adding $FeCl_3$ to M9. There was also a small but noticeable increase (~ 2-fold) across the *isc* genes, encoding enzymes for Fe-S cluster assembly, for cells grown in M9.

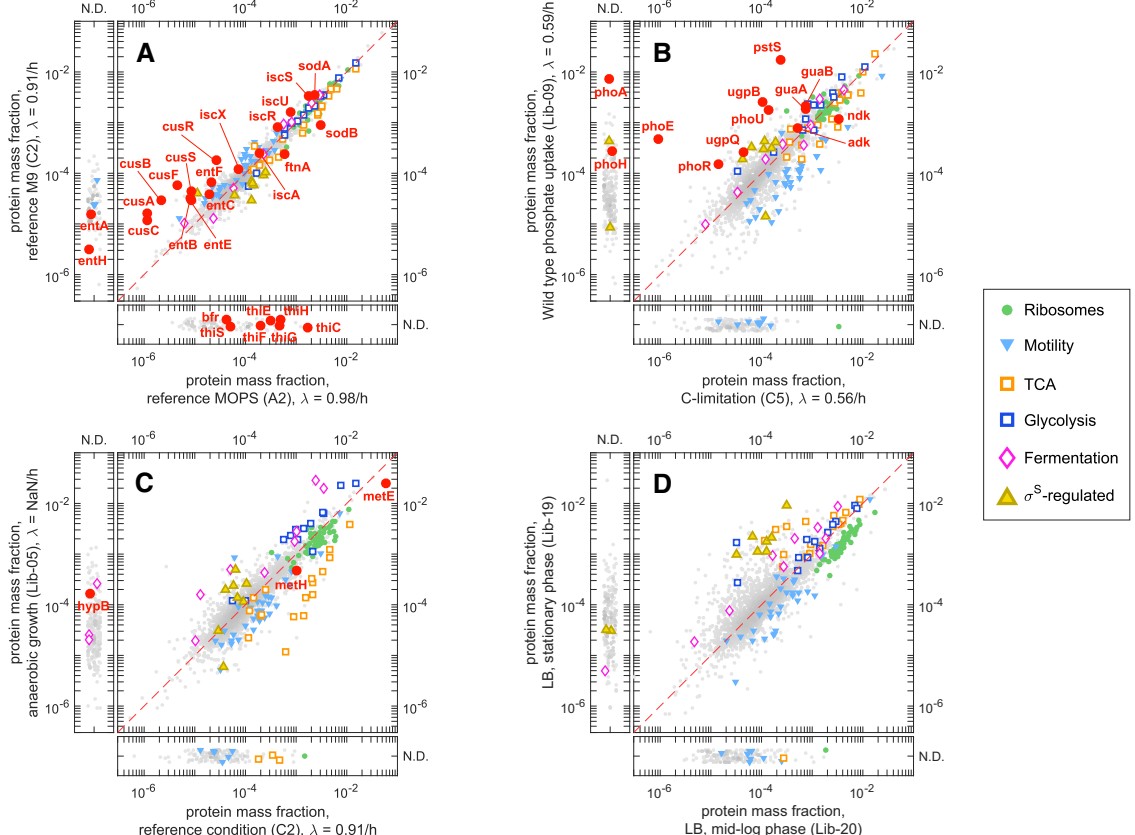

**Figure 7.  Comparison of proteome profiles across growth conditions.**

Scatter plots of absolute protein mass fractions between pairs of growth conditions and/or strains. Lateral boxes include proteins detected in only one of the two samples. Growth rates are reported on the axis labels when available. Colored symbols indicate groups of proteins as described in the legend (see Appendix Fig S10 for more details). Some notable proteins are highlighted in red.

A  Comparison of protein mass fractions for cells grown in different base media: M9+glucose (*y*-axis) versus MOPS+glucose (*x*-axis).

B  Protein mass fractions for *Escherichia coli* strain NQ1431 (*phnE+*) growing in glucose and a poor phosphate source (2-aminoethylphosphonate) versus wild-type NCM3722 cells in reference condition (glucose minimal medium with phosphate as P-source).

C  Proteome of anaerobically growing cells on glucose minimal medium versus that of cells in reference condition.

D  Comparison of proteome of cells in rich (LB) medium in different growth phases: mid-exponential ($OD_{600}$ = 0.6, *x*-axis) and stationary phase ($OD_{600}$ = 3.34, *y*-axis).

Concomitantly, the abundance of TCA enzymes (open orange squares) was reduced by about 20% in M9 media. Together, these data depict a quantitative picture of how M9-grown cells cope with iron limitation by increasing Fe-S assembly and reducing their major "consumers", the TCA enzymes. Finally, we noted in M9 medium a strong upregulation of the *cusABCF* genes, encoding the silver/copper detoxification system, as well as of *cusRS* encoding for the regulators of the cusABCF operon. The cause of this increase is not known and may be related to the iron shortage. These data suggest that MOPS is a superior medium for vegetative growth, even though the differences in overall gene expression and growth rate between MOPS and M9 appear to be small.

### Phosphate-limited growth

We next turned to other modes of nutrient limitation. The steady-state response of the proteome to phosphate limitation had been characterized using a medium with phosphonate as the sole P-source. Due to a nonsense mutation of *phnE* (encoding a subunit of the phosphonate ABC transporter) in *E. coli* K-12, we created strain NQ1431 which is isogenic to NCM3722 except for having a *phnE+* allele. NQ1431 grew exponentially at a rate of 0.59/h in MOPS glucose medium with phosphonate as the sole P-source. We compared the proteome of this P-limited culture to that of a C-limited culture that grew at a similar rate (strain NQ1243 in M9 glucose medium). As seen in Fig 7B, the P-limited culture showed a strongly increased abundance of the phosphate uptake system, including several *pho*, *ugp*, and *pst* genes encoding enzymes involved in the transport and scavenging of alternative sources of phosphorous (red circles in Fig 7B). The abundances of some of the general stress proteins (yellow upward triangles) also increased, while motility proteins (blue downward triangles) were reduced 3-fold. Interestingly, ribosomal proteins, the major consumer of cellular phosphate, remained nearly unchanged (green circles), as did the RNA polymerase components (not shown). The nucleotide biosynthesis pathways increased moderately, 1.5-fold on average, in particular the enzymes GMP synthetase (*guaA*, 2.4-fold increase) and IMP dehydrogenase (*guaB*, 2.9-fold increase). A notable exception to this trend was the 2.7-fold decrease in the abundance of the nucleoside diphosphate kinase (*ndk*), which was only partially compensated by a moderate (1.5-fold) increase in the expression of adenylate kinase (*adk*).

### Anaerobic growth

We next compared the proteome of NCM3722 cells growing exponentially in aerobic and anaerobic conditions in glucose minimal medium (Fig 7C). Anaerobic growth is expected to induce significant proteome rearrangements due to the much lower efficiency in energy generation. Indeed, we observed a total variation of about 25% of the proteome (Appendix Fig S11). However, the growth rates, 0.68/h and 0.91/h, respectively, were not very different. Ribosome content (green circles) also did not change much ($\approx$ 1.3 fold, comparable to the change in growth rate). As expected, the abundances of most TCA proteins (orange squares) were strongly reduced (~ 5-fold) in anaerobic conditions, by a total of 3% of the proteome. Glycolytic proteins (blue squares) and proteins associated with fermentation (magenta diamonds) increased in abundance by 2-fold (4% of proteome) and ~ 5-fold (5% of the proteome), respectively. Most notably, AdhE, PflB, Eno, and GapA took between 2

and 3% of the proteome mass each in anaerobic growth. HypB, involved in the maturation of the hydrogenases, was detected in anaerobic but not aerobic condition. The abundance of motility proteins (blue downward triangles) was reduced by about 2-fold (~ 1% of proteome). MetE, catalyzing the last step of the methionine biosynthesis pathway and being the most abundant proteins (6% of proteome) for growth in minimal medium, was also strongly downregulated (2.5% of proteome in anaerobic growth). However, this was not a result of anaerobic growth per se, as the medium used for anaerobic growth included vitamin $B_{12}$, which enables *E. coli* to replace MetE by the much more efficient MetH.

### Transition to stationary phase

The transition from exponential growth to stationary phase also had profound impact on the proteome. We grew *E. coli* NCM3722 cells in rich (LB) media and collected samples in mid-log phase ($OD_{600} = 0.6$) and in stationary phase ($OD_{600} = 3.34$). The corresponding proteomes are plotted against each other in Fig 7D. The abundances of ribosomal proteins (green) and motility proteins (downward blue triangles) in stationary phase were reduced 2- to 3-fold compared to log phase. At the same time, the abundances of stress proteins (yellow triangles) and some TCA proteins (orange squares) were strongly elevated (10- to 30-fold), as well as proteins associated with fermentation (magenta diamonds, 2.5-fold). Overall, this change involved a remodeling of > 30% of the proteome (Appendix Fig S11), which likely took place during the time these cells gradually slowed down in growth as various nutrient elements in the LB medium got exhausted. Such extensive reallocation of the proteome was also clearly visible in the abundance of the proteins sectors (Appendix Fig S12). Fast growing cells (early log, filled triangle in Appendix Fig S12) were characterized by a very large R-sector (~ 40% of total protein mass), due to the large expression of ribosomal proteins and associated factors, and a very small S-sector. Vice versa, cells entering stationary phase displayed a greatly reduced R-sector (~ 20% of the proteome) compared to early log cells. Instead, a considerable increase in S-sector proteins ($t$ ~ 25% of the proteome, to be compared to ~ 10% in fast growth) reflects the abundance of stress and catabolic proteins in this condition. Biosynthetic enzymes, belonging predominantly to the A- and U-sectors and not needed during growth on rich media, were expressed at low levels across all growth phases (~ 15% for the sum of the sectors).

### Different stress sources

After describing the response of the *E. coli* proteome to several nutrient conditions, we investigated its response to several sources of stress, including high temperature, hyperosmolarity, and oxidative stress.

### High temperature

First, we compared the proteome of NCM3722 cells grown exponentially in MOPS minimal medium with glucose at 42 and 37°C. Growth rates in both cases were similar (0.88/h versus 0.98/h), and the two proteomes were similar as well (only ~ 15% of total variation, Appendix Fig S11). The abundances of heat shock proteins and chaperones (shown in red in Fig 8A) were only mildly increased (< 3-fold increase at 42°C). Motility proteins (filled blue) changed the most, becoming nearly non-detectable at 42°C. This change is likely post-transcriptional in origin (De Lay & Gottesman, 2012),

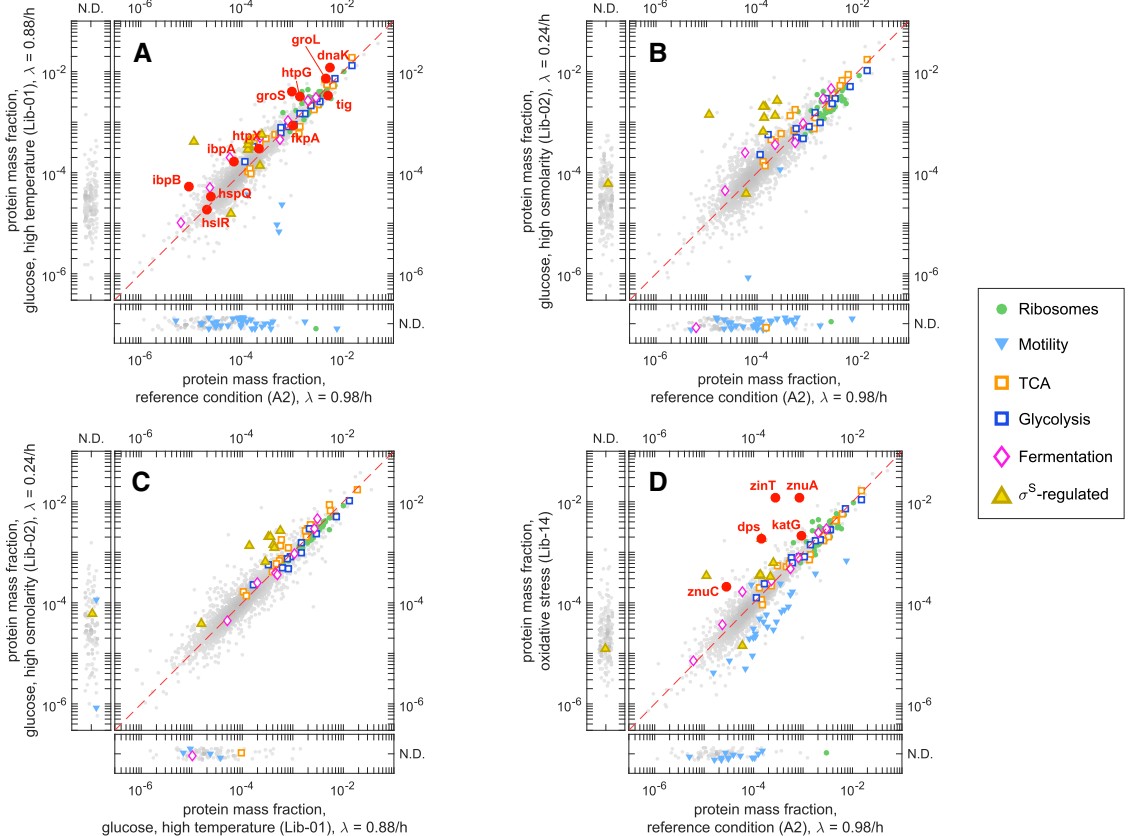

**Figure 8. Proteomes of cells subject to several stress conditions.**

Scatter plots of absolute protein mass fractions between pairs of growth conditions and/or strains. Symbols as indicated in the legend (same as in Fig 7); some notable proteins are highlighted in red.

A   Absolute protein mass fractions of NCM3722 cells grown at high temperature (42°C) versus normal temperature (37°C).

B   High osmolarity (0.6 M NaCl) versus normal osmolarity (0.1 M NaCl).

C   Comparison between proteomes of cells in high osmolarity versus high temperature. The two proteomes are quite similar despite the vastly different growth rates (0.24/h and 0.88/h, respectively).

D   Proteomes of cells subject to oxidative stress (600 μM hydrogen peroxide for 20 min before collection) compared to exponentially growing cells in reference condition.

since the regulatory region of *flhDC* encoding the motility master regulator contains a transposable element (Brown & Jun, 2015), which would relieve most transcriptional regulatory processes (Lyons *et al*, 2011; Cremer *et al*, 2019).

*High osmolarity*

We then compared the proteome of NCM3722 cells growing exponentially in MOPS glucose medium, between 0.6 M NaCl and normal osmolarity of 0.1 M NaCl (Fig 8B). Growth rates differed by almost 4-fold, and we observed an overall difference in proteome of about 25% (Appendix Fig S11). However, ribosomal proteins decreased by only 20% (from 15.4 to 12.3% of the proteome mass) while carbon-limited *E. coli* with similar reduction in growth rate experienced a 40% reduction (down to 9.3% of the proteome). This result can be attributed to the reported slowdown in translation for cells grown at high osmolarity, which led to compensatory increase in ribosome content (Dai *et al*, 2016). Compared to normal osmolarity, $\sigma^S$-dependent genes encoding general stress proteins did increase significantly (yellow upward triangles; 10-fold median increase),

while flagella (filled blue triangles) were again strongly reduced (1% change in proteome). It is instructive to compare the high osmolarity and the high-temperature samples (Fig 8C). The remarkable similarity of the two proteomes (< 20% total variation between the proteomes, Appendix Fig S11) suggests that the slowdown of growth in high osmolarity is not due to a protein allocation bottleneck, but may instead have a metabolic origin.

*Oxidative stress*

Finally, we looked at changes in the proteome of exponentially growing NCM3722 cells subjected abruptly to oxidative stress, i.e., 20 min after addition of 400 μM $H_2O_2$. Within this time span, we observed increased expression of several stress proteins (Fig 8D), including some known to be part of the oxidative stress response system (Dukan & Touati, 1996): Dps, KatG, ZinT, ZnuA, and ZnuC (red circles). Also notable is the rapid drop in the abundance of motility proteins, by more than 3-fold in 20-min (1.8% of proteome). A similar drop (at rate much faster than allowed by dilution due to cell growth) was observed also during nutrient

downshift (Erickson *et al*, 2017), suggesting an active mechanism of flagella loss.

### Various genotypes

In the last part of our analysis, we compared several *E. coli* genotypes different from NCM3722 and uncovered some surprising effects important to common laboratory studies.

### MG1655 vs NCM3722

First, we compared NCM3722 to MG1655 (sub-strain EQ353), both grown exponentially in MOPS glucose media (Fig 9A). EQ353 cells grow at a substantially slower rate compared to NCM3722 (0.69/h vs 0.98/h), which is consistent with the general trend that the growth of MG1655 cells being ~ 2/3 that of NCM3722 for a variety of carbon sources (Mori *et al*, 2016). Despite the difference in growth rates, the total difference in protein fractions was not very large, below 20% (Appendix Fig S11). Notably, we observed a strong reduction in the expression of motility genes, which are

mostly undetected in EQ353. This reduction can be attributed to the genotype. This particular sub-strain of MG1655 has a wild-type *flhDC* promoter (Cremer *et al*, 2019), whereas NCM3722 has the above-mentioned insertion of a transposable element immediately upstream of the *flhDC* promoter (Lyons *et al*, 2011), resulting in the constitutive expression of the FlhDC master motility regulator which is otherwise repressed (Prüss & Matsumura, 1997; Cremer *et al*, 2019). The strong (~ 25-fold) reduction in the expression of AceAB (enzymes of the glyoxylate shunt) in NCM3722 compared to MG1655 is also attributable to a known lesion: Reduction in AceAB is accompanied by a 3.5-fold increase in IclR, which is a key repressor of *aceAB* expression (Maloy & Nunn, 1982), while increase in IclR expression was predicted based on a mutation in the auto-regulatory region of *iclR* in NCM3722 (Lyons *et al*, 2011). A striking difference between MG1655 and NCM3722 was in the expression of outer membrane porins. OmpC was strongly expressed in EQ353 compared to NCM3722 (> 10-fold, 1.5% of proteome). The opposite behavior was observed for the other porin NmpC, which was very

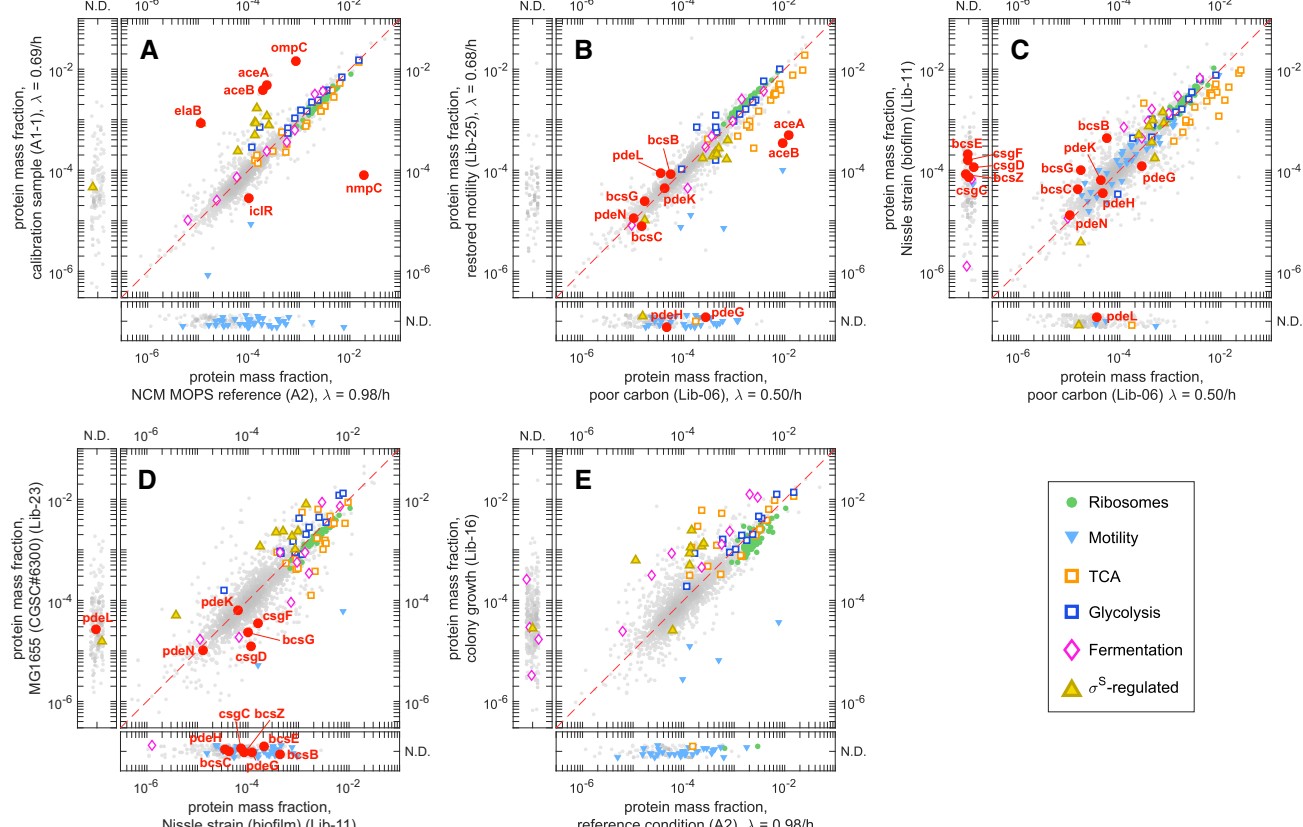

**Figure 9. Comparison of strains with different motility and biofilm capabilities.**

Scatter plots of absolute protein mass fractions between pairs of growth conditions and/or strains. Symbols as indicated in the legend (same as in Fig 7); some notable proteins are highlighted in red.

A   Absolute protein mass fractions of *Escherichia coli* MG1655 (EQ353) versus strain NCM3722, both growing on glucose minimal medium.

B   Proteome of the NCM3722-derived non-motile strain EQ59 (*motA⁻* with wild-type *flhDC* promoter) growing on melibiose, compared to that of NCM3722 cells growing on acetate.

C   Proteome of the biofilm-forming *E. coli* strain Nissle1917 compared to that of NCM3722 (both growing on acetate).

D   Comparison of *E. coli* MG1655 (Genetic Coli Stock Center strain #6300) and Nissle1917.

E   The proteome of *E. coli* strain EQ59 growing on solid agar is compared to that of NCM3722 in reference condition.

abundant in NCM3722 (~ 2% of proteome), 200× higher than that in EQ353. This difference is also attributable to the genotype: The *nmpC* gene in the genome of MG1655 is interrupted by an insertion element and is hence not functional. Appendix Fig S13A shows a comparison of the abundances of all porins in NCM3722 and EQ353. Apparently, the elevated level of OmpC and, to a lesser extent, OmpA, in EQ353, are a compensatory response to the loss of *nmpC*.

Finally, we also observed a general, modest upregulation of genes driven by the general stress sigma factor $\sigma^S$ (Fig 9A, yellow triangles), except for ElaB whose abundance was increased 75-fold (red circle), reaching 0.1% of the proteome in EQ353. The increased expression of $\sigma^S$-driven genes could arise from at least two scenarios. First, it could arise from genotype difference, as the *rpoS* gene in NCM3722 contains an amber stop codon which reduces its level of $\sigma^S$. However, direct comparison to a strain with restored *rpoS* in NCM3722 showed only moderate increase in $\sigma^S$-driven genes (Appendix Fig S13B), suggesting that the effect of *rpoS* mutation in NCM3722 might be minimal. Another possible scenario is sigma factor competition between $\sigma^S$ and the dedicated flagella sigma factor FliA (Mauri & Klumpp, 2014). In this scenario, the reduction in flagella expression (with undetectable FliA level) in EQ353 may favor the expression of $\sigma^S$-driven genes. The cause of the exceptional increase of ElaB level is not known.

### Motility mutant

We next compared NCM3722 to strain EQ59, which although nominally just *motA-*, has a restored wild-type *flhDC* promoter, as verified by sequencing (see Methods). The *flhDC* region of strain BW25113 (the ancestral strain from which the KEIO deletion library was derived (Baba *et al*, 2006)) does not contain any insertion elements (Grenier *et al*, 2014). Hence, the wild-type *flhDC* promoter was cotransduced, because it was adjacent to the *motA* null mutation. Comparing the proteome of EQ59 and NCM3722 (grown in carbon sources giving similar growth rates), we obtained a surprising finding. Fig 9B shows that EQ59 has, as expected, a strongly reduced expression of motility proteins compared to NCM3722 (filled blue triangles), with most motility proteins below the detection limit. Surprisingly, we also observed significant changes in the abundances of several *pde* genes, encoding phosphodiesterases (PDEs), which hydrolyze the signaling molecule cyclic-di-GMP coordinating biofilm formation (Reinders *et al*, 2016). In particular, PdeG and PdeH, the two most expressed PDEs in NCM3722, dropped to undetectable levels in EQ59, where PdeL and PdeK became the most dominant PDEs. The difference in PdeH expression can be linked to the mutation affecting the expression of motility proteins, since *pdeH* is predicted to be driven by a FliA-dependent promoter (Zhao *et al*, 2007), and this regulatory effect can be rationalized as necessary to suppress biofilm formation when the cell desires to turn on motility. We note that PdeH is commonly noted as the major PDE degrading cyclic-di-GMP, and is highly expressed in various *E. coli* strains (Pesavento *et al*, 2008; Reinders *et al*, 2016). The crosstalk between flagella and PdeH expression suggests that the high PdeH level seen in many laboratory strains may be an artifact of constitutive FlhDC expression. It is even conceivable that the biofilm-null phenotype is a key driving force favoring the motility+ phenotype in domesticated laboratory strains. The switch from the dominance of PdeH to PdeL in WT FlhDC

background is possibly very significant for the decision of biofilm formation in *E. coli*. PdeL is a complex molecule with enzymatic and regulatory activities, and its expression is auto-regulated (Reinders *et al*, 2016). The latter suggests an intriguing scenario where PdeL expression/activity may, through auto-regulation, set a switch-like decision for a cell's participation in biofilm formation. However, it should be noted that despite the low levels of PdeG and PdeH, EQ59 does not form biofilm due to other lesions common to domesticated K-12 strains (e.g., nonsense mutation in *bcsQ*, the first gene in the operon encoding cellulose production (Serra *et al*, 2013b)).

### Biofilm-former

To further study the expression of biofilm-associated genes, we analyzed protein expression in *E. coli* strain Nissle1917 (subsequently called Nissle). The Nissle strain is a non-pathogenic, non-K-12 strain commonly used as probiotic, which compared to K-12 strains such as NCM3722 is thought to have retained some wild-type characteristics, including the capacity to form biofilm (Bury *et al*, 2018). Indeed, strong clumping was observed when we grew Nissle cells in acetate in biofilm-forming condition (low osmolarity minimal carbon media into stationary phase, see Methods). Compared to NCM3722 cells grown exponentially in the same carbon source (Fig 9C), Nissle cells had similar levels of ribosomal, flagellar, and glycolytic proteins, but increased levels of fermentation enzymes (magenta diamonds) and lower levels of TCA proteins (orange squares). Significant changes were seen in the curli-associated proteins CsgC, CsgF, and the master regulator CsgD, which were expressed in Nissle cells but not in NCM3722 (red symbols on left side plot). Also, we observed an increase in the expression of Bcs proteins associated with cellulose production (red circles, > 3-fold). Interestingly, the phosphodiesterases PdeG and PdeH, highly expressed in the NCM3722 strain due to the constitutive motility expression mutation (see above), were still expressed in Nissle, but at somewhat lower level. PdeG abundance is reduced by 2-fold, which is presumably related to motility expression in Nissle. However, in Nissle the phosphodiesterase PdeL was not detected, suggesting that PdeL is in the "low" state and this strain is poised for biofilm formation in a different way from K-12.

MG1655 cells have different sub-strains which have or do not have insertion elements upstream of the *flhDC* promoter (Lee & Park, 2013), which are likely to affect their biofilm-producing capabilities. Here we took a strain without insertion element (CGSC#6300) and grew it in condition similar to that of the Nissle strain described above. When comparing its proteome to that of Nissle (Fig 9D), we observed that cellulose synthesis genes *bcsBCZ* were not expressed in MG1655 while product of *bcsG* was detected at a reduced level. This is likely a result of the aforementioned nonsense mutation in MG1655 that disrupted the *bcsQABZC* operon, but not the *bcsEFG* operon (Serra *et al*, 2013a). Also, Curli synthesis (Csg proteins) was detected in MG1655, but at a reduced level. Note that Curli synthesis proteins were not detected at all in NCM3722 and derivatives, likely due to defective Curli transport result from a nonsense mutation in *csgG* in NCM3722 (Lyons *et al*, 2011). Interestingly again, PdeL was detected in the non-biofilm-forming MG1655 strain, but not in the biofilm-forming Nissle strain.

We finally studied expression of the non-motile strain EQ59 on hard agar, where individual cells grow into compact colonies, with the radius of the colony expanding linearly rather than exponentially in time (Warren *et al*, 2019). The slow growth is attributed to non-uniform nutrient distribution within the colony, which caused cells in the interior of the colony to be in nutrient-starved state (Warren *et al,* 2019). In Fig 9E, we compared the (average) proteome of cells collected from such colonies versus exponentially grown planktonic cells, both in glucose minimal medium. Expectedly, motility proteins (filled blue triangles) were greatly reduced for the non-motile strain EQ59. Stress proteins (yellow triangles) were elevated in colony growth (on average ~ 10-fold). This is not surprising given that most cells in the colony are not growing due to the lack of nutrient, and are thus expected to resemble the stationary phase. However, ribosomal proteins were only moderately reduced, by ~ 60%, despite the very different growth behavior. Interestingly, the abundance of TCA genes (open blue squares) remained nearly unchanged, while those of fermentation genes (cyan) increased strongly (6-fold on average). This varied composition, resembling a mixture of anaerobic and aerobic growth, suggests varied micro-environments experienced by the cells, e.g., aerobic environment at the periphery and surface of the colony, and anaerobically in the colony interior.

# Discussion

The long-term goal of systems biology is the generation of predicative mathematical models to describe and understand cellular processes and phenotypes based on molecular data. For this goal, accurate, reproducible, and consistent measurements of absolute analyte concentrations across many conditions are crucial. Among the various biomolecules of a cell, proteins play a pivotal role, because they are actively involved in almost all cellular processes and many important biochemical events cannot be inferred from genomic or transcriptomic measurement (Liu *et al*, 2016). Recent advances in the field of proteomics, like DIA/SWATH mass spectrometry in combination with novel data analysis strategies, have matured in a way that it is possible to obtain accurate measurements of thousands of proteins reproducibly over many samples. However, obtaining accurate absolute abundances for lowly expressed proteins is still a challenge. Currently, biological insights derived from quantitative proteomic studies of *E. coli* have come mostly from coarse-grained levels, e.g., on the total abundances of enzymes participating in a metabolic pathway (Hui *et al*, 2015; Schmidt *et al*, 2016), but not from individual proteins.

In this paper, we developed a versatile workflow to obtain absolute abundances for thousands of proteins at the individual protein level over a vast number of conditions. DIA/SWATH mass spectrometry was used to generate high-quality peptide precursor intensity profiles at high throughput and minimal costs. For the peptide-centric DIA/SWATH data analysis, a comprehensive *E. coli* spectral library including 64% of all annotated *E. coli* proteins (2,770 proteins) was generated. A novel protein inference algorithm, termed xTop, was used to accurately infer protein intensities from peptide precursor intensities. xTop takes into account all detected peptide precursors for a given protein across all samples, thereby reducing the effect of noise and missing values in the dataset, and

producing better relative quantification compared to commonly used inference methods such as iBAQ and TopPep1/3 (Fig 2).

To obtain absolute protein mass fractions, we assessed various quantification methods using a set of calibration samples involving biological and technical replicates from the same condition. The absolute mass fractions obtained from proteomics data with various inference algorithms and from ribosome profiling data were evaluated using (i) a set of anchor proteins for which accurate protein abundances were measured with stable isotope-labeled synthetic peptides (AQUA), and (ii) several protein complexes with known stoichiometry. Among these methods, ribosome profiling yielded the most accurate quantitation, at least for proteomes which are not significantly degraded compared to dilution by growth rate (Koch & Levy, 1955; Mandelstam, 1958). As it is infeasible to use ribosome profiling to examine many samples in parallel, we used ribosome profiling as the standard for measuring absolute protein mass fractions and combined it with accurate relative quantitation over many samples offered by DIA/SWATH and the xTop protein inference method.

The resulting versatile workflow was used to quantify the absolute abundance of 2,335 proteins for various *E. coli* strains grown across 66 conditions, including stress conditions and non-planktonic states never characterized previously. Absolute abundances are reported in terms of protein mass fractions, which corresponds directly to cellular protein concentrations, regardless of the possible changes in cell size across growth conditions (Appendix Note S1). Our quantitative dataset includes low-abundant proteins with cellular protein mass fractions as small as $10^{-5}$ of the proteome, corresponding to concentrations close to $30/\mu m^3$ for an average-sized protein (~ 60 copies per cell for cells growing in glucose minimal medium), as well as high abundant proteins with cellular protein mass fractions close to 10% ($300,000/\mu m^3$, ~ 600,000 copies per cell).

We compared the results of this study to previous studies on the proteome of *E. coli*. Peebo *et al* (2015) studied *E. coli* BW25113 grown in chemostat (Peebo *et al*, 2015). We compared their results obtained in glucose minimal medium to ours for MG1655 (EQ353) grown in glucose batch culture. Their study showed a lower protein coverage (about 60% of ours). For the detected (more highly expressed) proteins, their absolute abundances were in good agreement with ours (Appendix Fig S14A and blue symbols in Appendix Fig S14D). Schmidt *et al* (2016) studied mostly BW25113 in a variety of batch and continuous cultures, but also provided data for MG1655 in glucose medium for comparison to other studies (Schmidt *et al*, 2016). We compared the latter to our results for MG1655 (EQ353) in glucose match culture. While the number of detected proteins were comparable (1,901 vs 1,843), we observed a systematic underestimation of low-abundant proteins (Appendix Fig S14B) and a larger variance in the stoichiometry of known protein complexes compared to our data (light blue in Appendix Fig S14D). These results are consistent with the bias of the iBAQ method used in that work for inferring protein abundances as discussed earlier (Appendix Figs S4D and S5). Finally, Caglar *et al* (2017) reported protein abundances for *E. coli* B strain (REL606) in a number of conditions (Caglar *et al*, 2017). We compared their data for glucose minimal medium in exponential growth to the same dataset for MG1655 (EQ353), which grew at similar rates (0.69/h compared to 0.75/h for REL606) in glucose minimal media. While the number of reported proteins was similar, the observed protein abundances

showed considerably more scatter compared to the other studies (S14C, gray symbols in S14D).

It is worth noting that the biological results discussed in these studies were largely insensitive to the problems with low-abundant proteins, as they were based mostly on the behavior of protein groups or on individual proteins that are highly abundant. This can be most clearly seen in the study by Hui *et al* (2015), in which abundances of individual proteins were not even reported (Hui *et al*, 2015). The other studies reported abundances mostly at the level of pathways or functional groups, which are dominated by the abundant proteins. Rarely abundances of individual proteins are discussed. In contrast, the robust quantification of protein mass fractions across conditions offered by our workflow led us to many interesting biological scenarios as summarized below.

The biological results analyzed in this work are divided into two parts, at the coarse-grained level and the individual protein level. Building on previous work (Hui *et al*, 2015), we first analyzed the proteome of *E. coli* at a coarse-grained level, in response to three major types of growth limiting conditions (metabolic limitation and antibiotic inhibition, 29 samples in total). This resulted in the classification of 1,821 proteins into eight "proteome sectors", each consisting of proteins with similar response profiles across these conditions. Our results are very much in agreement with those of Hui *et al* (2015) for the proteins detected in both studies. However, with our workflow a much increased resolution on low-abundant proteins was achieved, i.e., we quantified and classified almost twice as many proteins, although the absolute mass fractions of these proteins amounted to only ~ 10% more than those reported in Hui *et al* (2015). A large number of the newly detected proteins were upregulated in all three limitations (S'-sector), a pattern which was not seen previously. As many of these proteins are associated with cellular processes such as lipid and cell wall biosynthesis, cell division, cell cycle, and DNA replication, we attributed the increase in mass fraction of these proteins at slow growth to changes in the ratio of cell surface and cell number to cell volume, as cell volume decreases at slow growth (Si *et al*, 2017). This could be shown explicitly for outer membrane porins (Fig 5E), where the abundance per mass increased at slow growth, matching S/S'-sector profiles, whereas the surface density showed an approximate constancy across conditions.

In this study, we investigated an unprecedented variety of growth conditions, which allowed us to generate a comprehensive *E. coli* spectral library and provided us with a unique perspective to study *E. coli* physiology. Here we recap a few highlights. For example, the proteomes of cells in glucose-limited growth via titratable glucose uptake could be compared to those of cells growing on a variety of carbon sources. This yielded a unique perspective on protein allocation in carbon-limited growth (Fig 6). Surprisingly, only a small fraction of the proteins upregulated in carbon-limited growth were specific to the utilization of the supplied carbon sources. Half of these upregulated proteins were TCA enzymes and flagella-related proteins, while the rest were distributed between other generic transporters and the outer membrane porin NmpC. This result strongly challenges the view that proteome composition is optimally allocated for maximizing growth (O'Brien *et al*, 2013; de Groot *et al*, 2020; Dourado & Lercher, 2020), since many of these upregulated proteins are not directly useful for growth in the supplied carbon source.

The proteome of cells under various nutrient limitations and antibiotic inhibition showed strong differences as quantified in the sector analysis (Fig 4) and also from direct scatter plots (Appendix Fig S10). Surprisingly under stress conditions, specifically at high temperature (42°C) or under hyperosmotic stress (0.6 M NaCl), the proteome did not change significantly at a global level, despite a 4-fold difference in growth rates compared to unstressed conditions or among different stress conditions (Fig 8, Appendix Fig S11). This suggests that, unlike metabolic limitations and antibiotic inhibition where the rate of growth is intimately related to constraint in the proteome (You *et al*, 2013), the growth defects associated with these stress responses are likely not due to constraint in the proteome, but may instead originate from problems that cannot be solved by reallocating the proteome.

Another surprising finding from our dataset is the distinct differences on protein mass fractions resulting from seemingly minor effects. For example, M9 and MOPS are two commonly used growth media for laboratory studies of *E. coli*. However, the comparison of their proteomes shows a variety of clear differences caused by differences in thiamine and iron availability (Fig 7A). Similarly, different laboratory strains of *E. coli*, even those labeled as common MG1655 strain, exhibit important biological differences (Fig 9), particularly concerning the expression of genes involved in the synthesis and degradation of the cyclic-di-GMP, an important messenger molecule directing biofilm formation (Jenal *et al*, 2017). This is traced to the strain-dependent expression of FlhDC (Barker *et al*, 2004; Fahrner & Berg, 2015), the master regulator of flagella synthesis, which also drives the expression of PdeH (Ko & Park, 2000), the major phosphodiesterase responsible for cyclic-di-GMP degradation. This finding underscores the importance of strain selection in the study of both motility and biofilm formation.

So far, our analyses covered only a small part of the proteins accurately quantified in this study. Perspectively, our data can serve as a rich resource for the systems biology community, in order to perform hypothesis testing, integration with other quantitative omics-resources, or classical biochemical studies, as well as genome-wide modeling of *E. coli* gene expression and metabolism. For example, when combined with RNA-seq data on the same strains and conditions, these data can be used to infer translational efficiency for thousands of genes across many conditions. We expect data of this high quality and robust at the level of individual proteins, to herald in a new era in quantitative systems biology.

# Materials and Methods

### Organisms and culture conditions

A detailed description of all *E. coli* strains and growth conditions used in this study can be found in the Extended Experimental Methods and in Datasets EV1–EV3.

### *Escherichia coli* spectral library generation

### Proteomic sample preparation

The proteomic sample preparation was performed using an optimized *E. coli* protocol described previously by Schmidt *et al* (2016). Briefly, *E. coli* cell pellets were lysed with 2% sodium deoxycholate,

ultrasonicated, and heated to 95°C. Proteins were reduced, alkylated and digested with LysC and trypsin (Glatter *et al*, 2012). The peptide mixtures were desalted, dried, and resuspended to a concentration of 0.5 μg/μl. To all peptide mixtures, the iRT peptide mix (Biognosys) was added directly before the MS-measurement. To increase proteome coverage, 33 μg of peptides from samples Lib1 to Lib30 were pooled and fractionated by off-gel electrophoresis (OGE) into 13 fractions.

### DDA mass spectrometry

LC-MS/MS runs in DDA mode were performed on a TripleTOF 5600 mass spectrometer (SCIEX) interfaced with an NanoLC Ultra 2D Plus HPLC system (Eksigent). Peptides were separated using a 120 min gradient from 2 to 35% buffer B (0.1% v/v formic acid, 90% v/v acetonitrile). The 20 most intense precursors were selected for fragmentation. For the generation of the *E. coli* spectral library, 53 DDA-based proteomic measurements were performed in total (see Dataset EV2).

### DDA data analysis

The generated dataset was searched using four different search engines in parallel: Comet (Eng *et al*, 2013), Myrimatch (Tabb *et al*, 2007), X!Tandem (Craig & Beavis, 2003), and OMSSA (Geer *et al*, 2004). The MS2 spectra were queried against a canonical *E. coli* proteome database downloaded from Uniprot and appended with the iRT peptides (Biognosys) and 9 control or antibiotic resistance proteins. The search results were further processed and analyzed through the Trans-Proteomic-Pipeline (Deutsch *et al*, 2010). The combined results were filtered at a 1% protein FDR using MAYU (Reiter *et al*, 2009).

### Generation of spectral library and peptide query parameters

A non-redundant consensus spectral library (Lam *et al*, 2008) was generated with SpectraST (Lam *et al*, 2007). The python script "spectrast2tsv" (https://pypi.python.org/pypi/msproteomicstools) was used to extract peptide query parameters from the spectral library. This script automatically extracted the six most abundant singly or doubly charged b- and y-ion fragments for each peptide precursor in the range between 350 to 2,000 m/z, excluding the precursor isolation window region. iRT peptides were used to generate normalized retention times for all peptides.

## Quantitative proteomics with DIA/SWATH

### DIA/SWATH mass spectrometry

Tryptic peptides were measured in SWATH mode on two TripleTOF 5600 mass spectrometers (Sciex), both interfaced with an Eksigent NanoLC Ultra 2D Plus HPLC system as described previously (Collins *et al*, 2017). Peptides were separated using a 60 min gradient from 2 to 35% buffer B (0.1% (v/v) formic acid, 90% (v/v) acetonitrile). A 64-variable window DIA scheme was applied, covering the precursor mass range of 400–1,200 m/z (for details see Extended Experimental Methods), with a total cycle time of ~ 3.45 s. Per MS injection 2 μg of protein amount was loaded onto the HPLC column. For a detailed overview of all samples measured by DIA/SWATH, see Datasets EV2 and EV3.

### DIA/SWATH data analysis

The DIA/SWATH data was analyzed using OpenSWATH (www.openswath.org) as described previously (Collins *et al*, 2017). We

changed the following parameter: m/z extraction windows = 50 ppm. To extract the data, we used our *E. coli* spectral library described before. PyProphet-cli, an extended version of PyProphet, optimally combined peptide query scores into a single discriminative score and estimated q-values using a semi-supervised algorithm (Rosenberger *et al*, 2017). To assign the weight of each Open-SWATH subscore, we used the set of peptide peak groups subsampled from every run with the ratio of 0.07. The software was run using the experiment-wide and global context with a fixed lambda of 0.8, and the results of the experiment-wide mode were filtered with a 1% protein and peptide false discovery rate according to the global mode analysis (Rosenberger *et al*, 2017). TRIC was applied to align extracted and scored peak groups across all the runs following the filtration steps (Rost *et al*, 2016). The resulting peptide-level quantitative data matrices are available in Datasets EV4 and EV5.

### Quantitative protein inference using TopPep1/3, iBAQ, and xTop

Peptide precursors were ranked based on their average intensity over all samples (missing values are set to 0). Subsequently, the $N$ ($N = 1, 2, 3, \ldots$) most intense peptide precursors per protein were selected and their intensities were summed to yield the protein intensity in each sample. TopPep1 signals were simply the intensities of the one most intense peptide per protein, and TopPep3 was obtained by summing the top 3 peptide precursor intensities (Top peptides were defined not per sample, but by average over all samples). iBAQ intensities were computed by adding the signal from all peptide precursors and dividing by the expected number of fully tryptic peptides computed with an in-house script. Finally, xTop is a Bayesian estimator that makes use of all the available peptide intensities to improve on the TopPep1 estimation and is described in detail in Appendix Note S2.

Absolute protein concentrations or protein copy numbers were converted to mass fractions and vice versa through simple transformations using the known protein masses (see Appendix Note S1). Since TopPep1, TopPep3, iBAQ, and xTop most naturally reflect protein copy numbers or concentrations, we multiplied the corresponding intensities by the known molecular weight of each protein and normalized the corresponding intensities to 1. The resulting mass fractions $\phi_i$ are reported in Appendix Figs S4 and S5 for a representative MG1655 sample. The same rescaling has been applied to the ribosome profiling synthesis rates in order to generate the mass fractions $\rho_i$. The final scaling of xTop protein intensities with the ribosome profiling data was performed as described in Appendix Note S1; a scaling factor 1 was assigned to proteins for which no proteomics data were available in the calibration samples.

### Ribosome profiling

Ribosome profiling was performed as described in detail previously (Li *et al*, 2014). Briefly, 250 mL of cell culture in steady-state exponential growth was rapidly filtered at $OD_{600} = 0.3$ by passing it through a nitrocellulose filter. Cell pellets were rapidly collected using a pre-warmed metal crumber, flash-frozen in liquid nitrogen, and combined with frozen droplets of lysis buffer. Cells and lysis buffer were pulverized by mixer milling under cryogenic conditions. Pulverized lysate was thawed on ice and clarified by centrifugation at 4°C, digested with MNase, and the resulting monosomes collected by fractionation following sucrose gradient centrifugation. Ribosome

protected mRNA fragments were size selected via gel purification, dephosphorylated at the 3' end and ligated to 5' adenylated DNA oligo. After reverse transcription, the single-stranded DNA was circularized, and PCR amplified. Trimmed footprint reads were aligned to the *E. coli* genome using Bowtie v1.0.1 (Langmead *et al*, 2009). Relative synthesis rates were computed as the mean ribosome footprints density over gene bodies, with small corrections for 5' to 3' ramp and internal Shine-Dalgarno-like sequences.

**Binary classification of the proteome**

Proteome sectors were defined with a procedure similar to the one employed in Hui *et al* (2015). Briefly, for each protein and for each growth limitation series (C-, A- and R-limitation) we first normalized the mass fraction so that they equal 1 in the reference condition, and then fitted a linear relation between the relative protein abundance $r_i$ and the growth rate $\lambda$, namely $r_i(\lambda) = r_{i,0} + s\lambda$. The sign of the slope yields the "response" of the protein within each growth limitation mode: a negative sign corresponds to upregulation (↑), and a positive sign corresponds to downregulation (↓). Each sector is identified by the combination of three responses; for instance, the C-sector includes proteins upregulated in C-limitation (indicated as $C^\uparrow$) and downregulated in A- and R-limitation ($A^\downarrow R^\downarrow$). Such binary classification was performed only for proteins for which three data points were available in each limitation. All other proteins were assigned to an "X-sector" of unclassified proteins.

# Data availability

The datasets and computer code produced in this study are available in the following databases:

- raw mass spectrometry files (DDA and DIA/SWATH), ProteomeXchange Consortium via the PRIDE partner repository: http://www.ebi.ac.uk/pride/archive/projects/PXD014948.
- the *E. coli* spectral library file in various formats, SWATHAtlas (www.SWATHAtlas.org): http://www.peptideatlas.org/PASS/PASS01421.
- targeted analysis of selected peptides from the DIA/SWATH data, performed with the Skyline software (MacLean *et al*, 2010), available on Panorama Public (Sharma *et al*, 2018): https://panoramaweb.org/Ecoli_DIA.url.
- ribosome profiling data, Gene Expression Omnibus: https://www.ncbi.nlm.nih.gov/geo/query/acc.cgi?acc=GSE139983.
- xTop protein intensity calculator, MATLAB package, https://gitlab.com/mm87/xtop. A Python version of the code is currently being developed, and will be made available at the same repository.

**Expanded View** for this article is available online.

## Acknowledgements

We would like to acknowledge members of Hwa laboratory (Markus Arnoldini, Jonas Cremer, Xiongfei Dai, David Erickson, Tomoya Honda, Igor Segota, Mya Warren, Manlu Zhu) for collecting the array of *E. coli* samples used in this work. We would like to thank Ching Chiek Koh for his help in performing the off-gel electrophoresis. TH acknowledges the hospitality of the Institute for Theoretical Sciences at ETH, which enabled the initiation of this work. MM, ZZ, HO, and TH were supported by the NIH through grants R01GM109069 and R01GM095903. ABE and RA were supported by the SystemsX.ch project TbX and the National Institutes of Health project Omics4TB Disease Progression (U19 AI106761). Further, RA was supported by the European Research Council (ERC-20140AdG 670821). JBL was supported by an HHMI international student research fellowship and an NSERC fellowship. GWL and JBL were supported by NIH R35GM124732, the Pew Biomedical Scholars Program, and the Smith Family Awards for Excellence in Biomedical Research. DSL was supported by NRF 2016R1A2B4013204. CL was supported by EPIC-XS, project number 823839, funded by the Horizon 2020 program of the European Union. Open Access funding enabled and organized by Projekt DEAL.

## Author contributions

MM performed all the bioinformatics analysis following peptide identification, invented the xTop algorithm. MM, TH, and DSL analyzed the quantitative proteomics data. CL acquired the DIA and DDA data, fractionated the samples for the library, generated the spectral library, and ran the OpenSWATH pipeline. ABE performed additional proteomics analysis. ZZ prepared the proteomics samples. HO collected a large fraction of the samples and curated the corresponding experimental methods. JBL and GWL performed the ribosome profiling experiments and provided protein synthesis rates. AS and BC provided assistance with proteomics methods. OS contributed to the initial design of the project. DSL provided an early analysis of the proteomics data. MM, TH, and CL wrote the manuscript. CL, RA, and TH discussed and supervised the work. All authors reviewed the manuscript.

## Conflict of interest

The authors declare that they have no conflict of interest.

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
