## [Review Process File · Molecular Systems Biology]

From coarse to fine: The absolute Escherichia coli proteome under diverse growth conditions

Christina Ludwig, Ruedi Aebersold, Olga T. Schubert, Ben Collins, Amir Banaei-Esfahani, Alexander Schmidt, Gene-Wei Li, Terence Hwa, Zhongge Zhang, Jean-Benoit Lalanne, Hiroyuki Okano, Matteo Mori, and Deok-Sun Lee

DOI: 10.15252/msb.20209536

Corresponding author(s): Christina Ludwig (tina.ludwig@tum.de), Ruedi Aebersold (aebersold@imsb.biol.ethz.ch), Terence Hwa (hwa@ucsd.edu)

Review Timeline:

Submission Date:	21st Feb 20
Editorial Decision:	7th Apr 20
Revision Received:	9th Feb 21
Editorial Decision:	9th Mar 21
Revision Received:	7th Apr 21
Accepted:	9th Apr 21

Editor: Maria Polychronidou

Transaction Report:

Thank you again for submitting your work to Molecular Systems Biology. We have now heard back from two of the three referees who agreed to evaluate your study. Unfortunately, after a series of reminders we have not managed to obtain a report from reviewer #2. In the interest of time, we have decided to proceed with making a decision based on the two available reports. Overall, the reviewers acknowledge that the study presents resources and methodology that may be relevant for future analyses. However, they mention that as it stands the study seems somewhat preliminary and they raise a series of concerns, which we would ask you to address in a major revision.

Without repeating all the points listed below, some of the more fundamental issues raised are the following:

- Reviewer #1 points out that as it stands, the resulting novel insights into E. coli physiology seem rather limited. Including some follow up analyses enhancing the biological aspect of the work would significantly increase its impact.
- The methodological advance over existing approaches needs to be better supported and explained.
- Reviewer #3 offers several constructive suggestions on how to improve the study and more convincingly support the advantages of xTOP.
- Reviewer #1 raises some issues regarding the Ribo-seq analyses, which would need to be addressed.

All other issues raised by the reviewers would need to be convincingly addressed. Please let me know in case you would like to discuss any of the issues raised by the reviewers.

On a more editorial level, we would ask you to address the following issue.

REFEREE REPORTS

Reviewer #1:

The paper by Mori et al. provides an optimized method based on DIS/SWATH proteomics, which enables generation of qualitative data on E. coli protein abundance. The approach is parallelizable as a high-throughput so that it allows for sampling proteomes at low costs. The authors also develop a new algorithm xTop, which is based on the largely used DIA/SWATH pipeline, and allows for precise quantification of the E. coli proteome. As compared to other algorithms (e.g. TopPep1, TopPep3, iBAQ) xTop shows incrementally better performance. The mathematical assessment of

the data is impressive achieving this improvement of the absolute quantification of the proteome and the paper is technically sound. However, I am afraid that I did not learn much about the *E. coli* physiology. The following questions came to my mind when reading the paper:

1. What is the biological question that has been addressed that goes beyond those addressed with marginally worse performing algorithms? What did I learn more?
2. What is the add-on of this manuscript that goes beyond the already published proteome quantifications? Two of the senior authors (Terry Hwa and Ruedi Aebershold) are coauthors of two published papers ref.8 in *Mol Syst Biol* 2015 and ref. 6 in *Nat Biotechnol* 2016, respectively, which use the same breadth of the data, i.e. quite similar number of growth conditions.

To me the paper has a great technical value and the authors should rephrase it as algorithm-developing manuscript. It would be also good to fairly acknowledge that the performance of the existing algorithms is not bad and rather suggest to the readers what would be the biological advance of having *r*-values improvement from 0.9853 to 0.988.

Having said that, another major point of criticism is the inclusion and comparisons with the ribosome profiling data sets. While ribosome profiling approach bears some benefits for determining translation yields and velocity (by assuming average initiation rates), the quantification of the proteome is based on quite many (some of them unsubstantiated) assumptions, as it does not capture stability and degradation of proteins. Those are definitely not of the same order, unlike what the authors assume. Furthermore, ribosome profiling is skewed towards high abundance transcripts in somewhat length-dependent manner (Suppl. Fig. S4), which also supports the view that it is not a proxy of protein concentration. Looking at the data analysis closer, thought this is scarcely described, one curation of the data is worrisome: '...with corrections for...internal Shine-Dalgarno like sequences' (p.16, l. 616-617). The internal Shine-Dalgarno like could not be reproduced by many other groups and seem to be an artifact of skewed data analysis, i.e. they are present only when reads longer than those representing genuinely translating ribosome are considered (see the publication by Rachel Green's laboratory PMID. 26776510 on reanalyzing the data from PMID: 22456704). Since the ribosome profiling is very tangentially presented in this manuscript, and the comparisons to it are not a real add-on to the technical question, given also the possible flaws in data analysis, it would be better (and much clearer) to leave out those comparisons and focus only on the proteomics data.

Reviewer #3:

Summary

Mori et al. first present a new summary metric for absolute protein abundance derived from LC-MS experiments (not particularly specific to SWATH-MS), called xTop, and then use the quantitation method as the basis of profiling the quantitative variation of the *E. coli* proteome across multiple growth conditions and inferring sub-proteomes that drive unique response to individual modes of cellular stress. The work also involved generation of an exhaustively large MS/MS spectral library for the *E. coli* proteome, which will be a useful resource for other proteomics laboratories performing similar proteomics experiments with SWATH-MS mode of scans. I personally lack the expertise to evaluate the article based on the observations made for *E. coli* biology, and as such I will be unable to comment on the novelty of systems-level characterization presented in the latter half of the article. Instead, I will offer some comments on the xTop method, and say a few things about potential improvement in demonstrating the advantage of xTop in the real application.

Major comments

- This is more of a rhetorical comment, but I have seen numerous examples in eukaryotic proteome

analysis that publicly available spectral libraries are not optimal for DIA-MS data extraction, in terms of number of peptides quantified and their CoVs computed over replicates. Most often, in-house libraries obtained from the lab's own instrumentation (with DDA scans) were winners in our experience, especially with respect to the retention time information as it is column dependent (let alone the peaks in the product ion spectra). I would argue that DDA analysis with a certain level of fractionation isn't considered expensive or laborious, as these resources can be generated over a few iterations until the data saturates for a given organism and sample type. I am not rejecting the notion that the authors' spectral library is a good benchmark resource for future investigations, but its presence should not discourage individual laboratories from building their own spectral libraries - perhaps a sentence or two on this matter will serve as a good precautionary note.

- I read the Supplementary Material on xTop, especially Note S2. At the end of the day, what is the unit of absolute abundance in xTop? I can see the logic in the calculations from protein mass fractions to concentrations is sound, but there is no validation of accuracy for this conversion (rather the note cites another paper by the first author, reference 43?). Even without reading the note, I believe any analytical chemist would ask you for the ultimate test: the comparison against concentrations from fully targeted assay data, for at least a few proteins picked from a wide range of abundance (with focus on low abundance proteins), or data from biochemical assays with specific concentration units attached to those proteins with concentration differences of certain factor in the same sample (10 fold between each, or something like that). Without such experimental validation, all these calculations would have to remain speculative for a large majority of proteins (I would have argued the same for Schwanhausser paper if I were the referee). If you successfully prove a high correlation between xTop's concentrations and concentrations obtained from targeted assay data for a few tens of proteins, whose range cover the realistic dynamic range of concentrations measurable from MS experiments, that would be a truly exciting story to report.
- Related to the comment above, even if the authors' calculations are accurate for the detectable proteome, I wonder whether we eventually have to acknowledge that absolute quantification measures in untargeted MS-based proteomics are absolute only to a factor of a certain constant, especially for low abundance proteins. A discussion for and against this conjecture will be intriguing for some readers.
- How is the correlation with Ribo-seq data a validation of absolute protein abundance? Ribosome protected mRNA fragment counts, normalized by the counts from the bulk RNA-seq data, can provide a measure of new synthesis rates, rather than concentration value per se, don't they? Is this possible because we're looking at a prokaryote?
- Lastly, how much of the overall conclusion and individual protein specific findings (membership to sectors, for instance) would have changed if you analyzed the same data in different growth conditions with iBAQ? Can you provide some comparison from the real analysis example, highlighting the advantage of xTop over iBAQ?

Specific comments on xTop

- What is the connection between ϵ_p to the Equation N1.3? In the likelihood shown in N1.4, the random parameters seem to be x and σ , but it wasn't clear what the index r iterates over. Overall, the description of the likelihood and prior was very confusing, without proper introduction of notation and with constant change in subscripts across the paragraphs.
- Also, it looks like the whole inference formulation is hierarchical Bayes - is that correct? How were the MAP estimates obtained? By sampling (e.g. Markov chain Monte Carlo?) or is there a closed-form solution? The prior elicitation process is described in page 15 and 16, but I found the entire writeup essentially anecdotal, lacking systematic evaluation through specification of multiple sets of priors and proper numerical evaluation.
- In Fig. N1.1, isn't the reason why xTop has the smallest variance because it is a weighted sum and each peptide's contribution to the whole sum is below 1 (ϵ_p), which in turn lowers the overall

absolute level? You should probably switch the comparison to coefficient of variations, rather than variance.

- Figure 2C, main text line 200: I don't think the data shows any evidence of claiming superiority of xTop over others - in fact, they all look quite good. But I do welcome another measure comparable to iBAQ - it's obvious that iBAQ is not panacea to all problems. Would it be possible that xTop may have advantage specifically in the low abundance area. I would speculate that, since iBAQ adds up any peptide regardless of consistent detection across samples, xTop's weighting tactic should work better in low abundance proteins with patchy detection rates?
- Another interesting thought: how is the overlap between every pair of the four approaches? Are they all well correlated? Can you provide a supplementary figure showing all pairwise plot? Such a plot would eventually reveal the area of abundance range where the four strategies differ.
- Can your MATLAB code run on freely distributed software environment (like Octave)?
- What do you think would happen if you were to apply xTop to an eukaryotic proteome? How do you think it will fare against iBAQ? (Disclaimer: I am not a fan of iBAQ at all).
- Lastly, I'd like to enquire the behavior of xTop in the following scenario, for a protein with three peptides, with samples with identification and ion chromatogram extraction indicated as o's (detected) and x's (not detected) across five samples:

Peptide S1 S2 S3 S4 S5

1 o o o x x

2 x o o o x

3 x x o o o

(1) Do you think we should quantify this protein at all?

(2) Can ϵ_p be robustly estimated by xTop, with this particular configuration of detection (barring all limitations stemming from a small sample size)?

(3) The goal of the experiment was to compare abundance between group A (S1 + S2) and group B (S4 + S5). Is such comparison feasible?

As you can already imagine, I am asking this question for one reason: I think you should consider incorporating a step where xTop declares certain proteins "not quantifiable". The criterion of retaining peptides detected in at least two samples is certainly not enough. The field of quantitative proteomics has always run towards one goal: quantifying more proteins. Almost no paper discusses the point of "quantifiable" fraction of proteome.

Reviewer #1:

The paper by Mori et al. provides an optimized method based on DIS/SWATH proteomics, which enables generation of qualitative data on E. coli protein abundance. The approach is parallelizable as a high-throughput so that it allows for sampling proteomes at low costs. The authors also develop a new algorithm xTop, which is based on the largely used DIA/SWATH pipeline, and allows for precise quantification of the E. coli proteome. As compared to other algorithms (e.g. TopPep1, TopPep3, iBAQ) xTop shows incrementally better performance. The mathematical assessment of the data is impressive achieving this improvement of the absolute quantification of the proteome and the paper is technically sound. However, I am afraid that I did not learn much about the E. coli physiology. The following questions came to my mind when reading the paper:

1. What is the biological question that has been addressed that goes beyond those addressed with marginally worse performing algorithms? What did I learn more?

We appreciate the Reviewer's acknowledgement of our effort. In the new version, we explained more clearly the key ingredients to our versatile workflow that led to the high throughput and low cost characterization of the absolute proteome: The main role of xTop was to provide an improved relative abundance characterization that is more robust (i.e. less noisy and less dependent on specific protein attributes such as the detection efficiency of its peptide precursors) than existing method, as detailed in Fig. 2 and Appendix Note S2. The other important ingredient is the use of ribosome profiling for absolute abundance determination for selected samples.

2. What is the add-on of this manuscript that goes beyond the already published proteome quantifications? Two of the senior authors (Terry Hwa and Ruedi Aebershold) are coauthors of two published papers ref.8 in Mol Syst Biol 2015 and ref. 6 in Nat Biotechnol 2016, respectively, which use the same breadth of the data, i.e. quite similar number of growth conditions.

The output of this study is the absolute abundance of > 2000 proteins down to a coverage of ca. 50 copies per cell, for over 60 growth conditions. We thank the reviewer for asking this

pointed question on what has been learned about the physiology of *E. coli*, which is the absolute yard stick for assessing the utility of a methodology. In response to the reviewer's question, we added a major new section on biological findings, which effectively doubled the length from the original submission. Salient summary of the findings is given below and in the Discussion section:

- **Expression of low abundant proteins.** A large number of low-abundant proteins are shown to increase under carbon limitation, anabolic limitation, and translational inhibition, as shown in Fig. 4 and Appendix Figures S8 and S9. This is the opposite of earlier findings by Hui et al. (1), which found abundant proteins to increase mostly under a single limitation and decrease under the other limitations. These proteins are numerous in number (comparable to the abundant proteins detected in Hui et al.) but small in their aggregated abundance (~10% of those in Hui et al.); the latter ensures that their behavior do not significantly affect the overall proteome allocation as described by Hui et al. However, the existence of a large number of proteins responding in ways different from the abundant proteins point to many effects left out of the existing models of proteome allocation by Scott et al. (2) and Hui et al. While these models focus exclusively on proteins responsible for biomass growth, the new behaviors found for the low-abundant proteins point to roles played by proteins whose activities affect cell surface (e.g., lipids, cell wall, periplasm) and cell number (e.g., DNA replication, cell pole and cell division). We expect that our data will stimulate the construction of more advanced models that take these effects into account.
- **Composition of catabolic proteins.** Because we investigated quite a number of samples with different carbon sources, we were able to compare the proteome of these samples to those with similar growth rates, but generated with titration of glucose uptake (see new Fig. 6 and Fig. 7). This comparison revealed differences in the composition of catabolic proteins which were upregulated generally under "carbon limitation". It turned out that most of these upregulated catabolic enzymes are common regardless of the carbon sources used; they include TCA enzymes, flagella proteins, and many generic transporters such as those for various amino acids and dipeptides, as well as for the uptake of acetate, ribose, and maltose, reflecting a picture of carbon foraging response. In fact, allocation of the catabolic enzymes dedicated to the specific sugar supplied in the growth medium comprise at most 10% of the entire pool of proteins upregulated in C-limitation. This finding is in contrast with claims of "optimal" proteome allocation to maximize cell growth as proposed by various theoretical studies (3-5). These new data analyses were added to the manuscript in the paragraphs "Different carbon sources", "Other nutrient conditions", "MOPS versus M9", "Phosphate-limited growth", "Anaerobic growth" and "Transition to stationary phase".
- **Proteome response to applied stress.** In contrast to large, proteome-scale reallocation as growth rate gets varied due to metabolic limitation and translational inhibition, proteome-scale reallocation was surprisingly mild in response to the stress we applied (see new Fig. 8). This was particularly clear for cells grown under high temperature (42 °C) and high osmolarity (0.6 M NaCl). Despite a 4-fold difference in growth rate, their proteome changed little from each other, and both were not so different from reference state grown in glucose minimal medium at 37 °C and normal

osmolarity. While these results need to be followed up further with detailed studies for each applied stress, we believe the surprising findings suggest an important concept, that growth defect due to these applied stresses cannot be compensated by adjusting the proteome. In other words, growth limitation occurring in these cases are not the result of proteome constraint as reported previously for metabolic and translation limitations. These new data analyses parts were added to the manuscript in sections “Different stress sources”, “High temperature”, “High osmolarity” and “Oxidative stress”. We will have a lot more to say about the nature of growth limitation in follow-up studies.

- **Strain-dependent effects.** We compared a number of commonly studied laboratory strains of *E. coli* and found some important differences (see new Fig. 9). Most striking were the differences between sub-strains of MG1655, where those with gratuitous motility expression (due to insertion element upstream of *flhDC* operon encoding the master motility regulator) were found to have very different expression of enzymes central to the synthesis and degradation of cyclic-di-GMP, the key signaling molecule controlling biofilm formation. We expect this finding to be important for resolving many of the conflicting reports regarding cyclic-di-GMP in *E. coli*. We have added the paragraphs “MG1655 vs NCM3722”, “Motility mutant” and “Biofilm-former” to the results of the revised manuscript.

To me the paper has a great technical value and the authors should rephrase it as algorithm-developing manuscript. It would be also good to fairly acknowledge that the performance of the existing algorithms is not bad and rather suggest to the readers what would be the biological advance of having r-values improvement from 0.9853 to 0.988.

Because the main contribution of this work lies in the estimated concentrations of low-abundant proteins, we feel our discussion of the biological findings, prompted by the reviewer’s question above, is central to establishing the technical advances. As stated above, no studies published so far have delved into biologically important issues based on the abundances of individual proteins.

Regarding the r-values: We agree with the reviewer that r-values were generally good for all methods tested here. For example, correlation coefficients between biological replicates were above 0.97 for all four proteomics methods tested (Appendix Fig. S2D). However, as for any descriptive statistics, the correlation coefficients only provide a partial perspective on the quality of the replicates, not the full picture. In this case, similarly high correlation coefficients for pairs of replicates were accompanied to strikingly different (up to 3-fold) variances for the log-ratio of the protein abundances. Indeed, one of our main points is that **despite** the good r-values, each method presents some systematic biases that affect accurate protein quantification at either relative (Fig. 2) or absolute level (Fig. 3, Appendix Fig. S4). These biases, which skew the estimated protein abundances by factors as large as 3-fold, are not easily captured by differences in the r-values. Instead, they are better quantified using different tools, such as the multivariate analysis presented in Appendix Note S3 (summarized in Fig. 3F).

In this new version of the manuscript we rarely rely on the Pearson correlation coefficient to quantify the “spread” between pairs of samples. Rather, we use variance of log-ratios (Fig. 2H, Fig. S2, Fig. 3F, Fig. S11), coefficients of variation (Fig. 2I-L), fraction of proteins with

discrepancy larger than 2-fold (Fig. 3K), or total variation in total mass fraction (Fig. S11). The only two cases in which we mention the Pearson correlation coefficient in the Main Text are the following:

1) When discussing the good correlation with AQUA-derived protein abundances for all four methods (Lines 237-239: “As shown in Appendix Fig. S3A-D, all four protein inference algorithms showed good correlations to the abundances determined using AQUA peptides ($r > 0.92$)”)

2) When discussing the differences in relative quantification between different methods (Lines 298-302 and Fig. 3G-J). In this case, the correlation coefficients span a broader range (0.778 and 0.865) compared to the previous cases, so the improvement of xTop over TopPep3 and iBAQ is more apparent. This result is further supported by the quantification (in Fig. 3K) of proteins deviating at least 2-fold from the diagonal, which shows striking differences between TopPep1/xTop and TopPep3/iBAQ.

Having said that, another major point of criticism is the inclusion and comparisons with the ribosome profiling data sets. ... Since the ribosome profiling is very tangentially presented in this manuscript, and the comparisons to it are not a real add-on to the technical question, given also the possible flaws in data analysis, it would be better (and much clearer) to leave out those comparisons and focus only on the proteomics data.

Ribosome profiling is an essential part of this work. We have added a section on absolute protein quantification (Lines 231-287), as well a dedicated section in Appendix Note S1 (“Ribosome profiling and absolute protein quantification”). The results in Fig. 3A-E show that absolute abundance estimates based on ribosome profiling result are superior compared to all the other methods tested. This conclusion is based on comparison to calibration by AQUA peptides (Fig. 3A) as well as by internal standard based on protein complexes with known protein stoichiometry (Fig. 3B-E). In all cases, ribosome profiling came out on top. This is obviously not an appealing result for mass spectrometry enthusiasts, and invoked quite a bit of discussion among our co-authors. But we finally came to the agreement that for proteome such as that of *E. coli* during exponential growth with negligible turnover, the advantage of ribosome profiling in its ability to characterize the absolute abundance of lowly expressed proteins is undeniable.

Responses to the reviewer’s concerns on the technical aspects of ribosome profiling are given below.

While ribosome profiling approach bears some benefits for determining translation yields and velocity (by assuming average initiation rates), the quantification of the proteome is based on quite many (some of them unsubstantiated) assumptions, as it does not capture stability and degradation of proteins. Those are definitely not of the same order, unlike what the authors assume.

We agree with the reviewer that stability and degradation of the proteins are not of the same order, a point that might not have been made sufficiently clear in the previous version of the

manuscript. Indeed, the quantification of absolute protein abundances via ribosome profiling is viable, because proteins are mostly stable over the cellular doubling timescale in exponentially growing *E. coli* cells. This is now discussed in detail in Appendix Note S1, section “Ribosome profiling and absolute protein quantification”, and highlighted in the Main Text (lines 249-251):

“Since protein degradation is negligible for the vast majority of proteins in exponentially growing *E. coli* cells (13, 28, 29), synthesis rates are proportional to absolute protein copy numbers. In turn, these can be converted to absolute protein mass fractions (Appendix Note S1).”

as well as in the Discussion (lines 792-794):

“Among these methods, ribosome profiling yielded the most accurate quantitation, at least for proteomes which are not significantly degraded compared to dilution by growth rate (28, 29).”

Other requirements for the use of ribosome profiling data for absolute protein quantification, e.g. that ribosome elongation speed is similar across different mRNA species, are also discussed in Appendix Note S1. For eukaryotic cells, growth and decay rates are known to be closer in magnitude, so ribosome profiling cannot be straightforwardly applied.

Furthermore, ribosome profiling is skewed towards high abundance transcripts in somewhat length-dependent manner (Suppl. Fig. S4), which also supports the view that it is not a proxy of protein concentration.

We are not sure what the reviewer is referring to, since Fig. 3G-J (previously Supp. Fig. Figure S4) does not display any information on protein length. In any case, we are not aware of any systematic biases of ribosome profiling with expression (e.g., no trends in a comparison between ribosome profiling and a curated list of protein abundances are reported in the literature, Fig. 1B of Li et al. (6)). We do agree that genes expressed at low levels will typically be more challenging to quantify due to low sequencing coverage, however, when comparing ribosome profiling data to mass spectrometry data, it is usually the latter having the smallest coverage.

Looking at the data analysis closer, thought this is scarcely described, one curation of the data is worrisome: ‘...with corrections for...internal Shine-Dalgarno like sequences’ (p.16, l. 616-617). The internal Shine-Dalgarno like could not be reproduced by many other groups and seem to be an artifact of skewed data analysis, i.e. they are present only when reads longer than those representing genuinely translating ribosome are considered (see the publication by Rachel Green's laboratory PMID. 26776510 on reanalyzing the data from PMID: 22456704).

The Shine-Dalgarno correction is very small for nearly all genes (the distribution of fold-changes before/after the correction across all genes has a 5th and 95th percentiles of 0.91

and 1.08) and has thus limited effects on the estimated synthesis rates. Hence, the correction does not affect any of our conclusions.

This correction is included to ensure that the local differences in elongation rates are accounted for. We note that the analysis from Rachel Green's lab uses 3' end mapping of ribosome footprints, which correlates well with the position of the decoding center but dilutes the signal from the Shine-Dalgarno sequence at a variable distance upstream. Nevertheless, the inclusion of the SD correction has limited effects on the elongation rate averaged over the entire gene body. Importantly, our ribosome profiling protocol has been capturing a wide range of ribosome footprint lengths (15 to 40 nt) since 2014. This is considered the gold standard from the Green's lab exhaustive assessment of bacterial ribosome profiling (7).

Reviewer #3:

Summary

Mori et al. first present a new summary metric for absolute protein abundance derived from LC-MS experiments (not particularly specific to SWATH-MS), called xTop, and then use the quantitation method as the basis of profiling the quantitative variation of the *E. coli* proteome across multiple growth conditions and inferring sub-proteomes that drive unique response to individual modes of cellular stress. The work also involved generation of an exhaustively large MS/MS spectral library for the *E. coli* proteome, which will be a useful resource for other proteomics laboratories performing similar proteomics experiments with SWATH-MS mode of scans. I personally lack the expertise to evaluate the article based on the observations made for *E. coli* biology, and as such I will be unable to comment on the novelty of systems-level characterization presented in the latter half of the article. Instead, I will offer some comments on the xTop method, and say a few things about potential improvement in demonstrating the advantage of xTop in the real application.

We appreciate the reviewer's candid and insightful comments on our work. The part on *E. coli* biology has been extensively expanded, partly in response to questions raised by Reviewer 1, and partly because we feel the best way to support the advantages of the new method developed, is to discuss biology that has not been addressed by previous methods. We point out that while previous work claimed the ability to quantify the absolute abundances of many proteins, none of the biological results discussed relied on the abundance of individual proteins. Indeed, this work is the first one that is able to discuss interesting biology issues based on the abundances of individual proteins. This is in itself a testimonial to our new methodology which significantly reduced noise and reproducibility issues to allow us to address changes in individual proteins under different growth conditions.

Major comments

- This is more of a rhetorical comment, but I have seen numerous examples in eukaryotic proteome analysis that publicly available spectral libraries are not optimal for DIA-MS data extraction, in terms of number of peptides quantified and their CoVs computed over replicates. Most often, in-house libraries obtained from the lab's own instrumentation (with DDA scans) were winners in our experience, especially with respect to the retention time information as it is column dependent (let alone the peaks in the product ion spectra). I would argue that DDA analysis with a certain level of fractionation isn't considered expensive or laborious, as these resources can be generated over a few iterations until the data saturates for a given organism and sample type. I am not rejecting the notion that the authors' spectral library is a good benchmark resource for future investigations, but its presence should not discourage individual laboratories from building their own spectral libraries - perhaps a sentence or two on this matter will serve as a good precautionary note.

In the new version of the manuscript we added a large plethora of new results, and therefore we feel that the new spectral library is less central to the main theme of the manuscript anymore. While we appreciate the reviewer's comment and fully agree to the arguments stated, we have significantly scaled down the text sections regarding the spectral library generation in both the Introduction and the Discussion.

- I read the Supplementary Material on xTop, especially Note S2. At the end of the day, what is the unit of absolute abundance in xTop? I can see the logic in the calculations from protein mass fractions to concentrations is sound, but there is no validation of accuracy for this conversion (rather the note cites another paper by the first author, reference 43?). Even without reading the note, I believe any analytical chemist would ask you for the ultimate test: the comparison against concentrations from fully targeted assay data, for at least a few proteins picked from a wide range of abundance (with focus on low abundance proteins), or data from biochemical assays with specific concentration units attached to those proteins with concentration differences of certain factor in the same sample (10 fold between each, or something like that). Without such experimental validation, all these calculations would have to remain speculative for a large majority of proteins (I would have argued the same for Schwanhausser paper if I were the referee). If you successfully prove a high correlation between xTop's concentrations and concentrations obtained from targeted assay data for a few tens of proteins, whose range cover the realistic dynamic range of concentrations measurable from MS experiments, that would be a truly exciting story to report.

We thank the Reviewer for his/her precious comment. First of all, we always report absolute protein abundances in units of mass fractions ϕ_i , the mass of i^{th} protein over the total protein mass in the sample, except when specified otherwise. In order to avoid any confusion, we have clarified what we mean by "relative abundance" and "absolute abundance" in the Introduction (lines 59-63):

"Both relative protein quantification (allowing cross-sample comparisons for the same protein) and absolute protein quantification (allowing cross-protein comparisons in the same sample) provide crucial information on the activity of biochemical and regulation pathways,

on the stoichiometry of protein complexes, or on the relationship between gene expression and cellular phenotype (4-6).”

The relations between protein intensities and protein mass fractions is also explained in more detail in Appendix Note S1. We hope that this clarified any confusion we might have inadvertently generated.

Prompted by the Reviewer, we have now added (Fig. 3A, Appendix Figure S3) a comparison between MS-based and ribosome profiling-based protein mass fractions to a set of proteins measured using stable-labeled peptides (AQUA peptides) spiked in our calibration samples. This provided an independent absolute measurement for 29 proteins whose mass fractions spanned more than two orders of magnitude. The mass fractions computed with all methods tested (xTop, TopPep1/3 and iBAQ, plus ribosome profiling) all compare well with the AQUA abundances (Appendix Fig. S3A-E), with correlation coefficients above 0.92 in all cases, supporting the protein mass fraction calculations for all methods. The case of ribosome profiling is perhaps the most striking: since ribosome profiling measures protein synthesis rates, not protein abundances, one would intuitively expect lower correlation with actual protein abundances. Instead, ribosome profiling-derived mass fractions are the ones most correlated to AQUA-based mass fractions ($r = 0.97$) among all methods tested. We underlined this result in lines 254-257.

“We found that ribosome profiling-derived mass fractions correlated with AQUA-measured proteins as strongly as those derived from iBAQ, with 50% of the genes within a 1.5-fold interval (Fig. 3A, blue bar and symbols; Appendix Fig. S3E). This suggests that ribosome profiling provides good absolute protein quantification.”

The second method that we used to determine which protein quantification method is most accurate was the quantification of protein abundances in protein complexes (Fig. 3B-E). We indeed found that ribosome profiling provided the most reliable method for absolute protein quantification, and we hence used it as the basis of our versatile workflow.

• Related to the comment above, even if the authors' calculations are accurate for the detectable proteome, I wonder whether we eventually have to acknowledge that absolute quantification measures in untargeted MS-based proteomics are absolute only to a factor of a certain constant, especially for low abundance proteins. A discussion for and against this conjecture will be intriguing for some readers.

The comparison between proteomics-based and ribosome profiling-based protein mass fractions provided indications that this might be true for xTop and TopPep1, but not for TopPep3 and iBAQ (Appendix Fig. S4 and Appendix Note S3). We hope that in this version of the manuscript this message comes across better.

As shown in Fig. 3G and 3J, relative quantification with xTop and TopPep1 was in good agreement with relative quantification obtained with ribosome profiling. However, when testing the absolute quantification of these methods, we found a dependence of the absolute protein abundances on the protein size (Fig. 3F, Appendix Fig. S4D-E), thus supporting that xTop/TopPep1 mass fractions are indeed absolute only up to a (protein-dependent) constant. In our workflow, we therefore “calibrated” the relative proteomics data with absolute ribosome profiling data, but other approaches are possible. We discuss in Appendix

Note S3 how it is possible to “rescue” the xTop and TopPep1 protein abundances by calibrating them with a protein size-dependent factor: this allows to remove the observed bias and thus improve on the absolute quantification.

However, not all MS-based protein quantification methods are amenable to the same treatment. iBAQ has the opposite characteristics compared to the previous case: it does not display biases for/against differently sized proteins, but is strongly affected by missing peptides (Fig. 2F, Fig. N2.1 in Appendix Note S2). This affects both the absolute and the relative quantification offered by this method, and therefore iBAQ mass fractions cannot be “rescued” by multiplication with a constant (protein-dependent) factor. Finally, TopPep3 combines the worst behaviors from the above methods, namely the protein-length bias and the sensitivity to missing peptides, especially for low-abundant proteins.

• How is the correlation with Ribo-seq data a validation of absolute protein abundance? Ribosome protected mRNA fragment counts, normalized by the counts from the bulk RNA-seq data, can provide a measure of new synthesis rates, rather than concentration value per se, don't they? Is this possible because we're looking at a prokaryote?

The short answer to *why* does ribosome profiling work *at all* as protein quantification method is that in fast-growing *E. coli* cells, protein degradation is negligible for most proteins. Therefore, for these proteins, the absolute protein synthesis rates quantified via ribosome profiling are proportional to absolute protein abundances. This is now stated clearly in the Results (lines 249-252) as well as in the Discussion (lines 792-794). (See also answers to reviewer #1)

Crucially, in this version of the manuscript, we independently validated the use of ribosome profiling for absolute protein quantification, at least for a subset of AQUA-quantified proteins and for protein complexes (Fig. 3A-E; see above). This supports the use of ribosome profiling as a genome-wide gold standard for absolute protein quantification.

Prompted by the questions of both Reviewers, we also introduced a new section in Appendix Note S1 (“Ribosome profiling and absolute protein quantification”) to describe in detail the conditions required for viable absolute protein quantification via ribosome profiling (which are all met in exponentially growing *E. coli* cells). As the Reviewer can imagine, the application to eukaryotic cells, or even slow-growing prokaryotic cells, is likely to be problematic due to relatively stronger protein degradation compared to dilution.

• Lastly, how much of the overall conclusion and individual protein specific findings (membership to sectors, for instance) would have changed if you analyzed the same data in different growth conditions with iBAQ? Can you provide some comparison from the real analysis example, highlighting the advantage of xTop over iBAQ?

The impact of missing peptides on iBAQ and TopPep3 is probably best illustrated in Fig. 2C-G: the abundance of the protein RseA varies five-fold across replicate samples, where protein abundances should be the same. This has a marked effect on the relative quantification of individual proteins. We illustrated these effects in Fig. N2.1 in Appendix Note 2, which we copy here below for convenience of the Reviewer:

In this figure we compared the abundance of peptide precursors (top row) and estimated protein abundances (bottom row) for three proteins (Sbp, YgeR, FadH) at different growth rates in carbon-limited conditions. As clearly seen in the case of Sbp and FadH, iBAQ (blue diamonds, proportional to the sum of all peptide intensities in the top panels) greatly overestimates the variation in protein abundances across conditions with respect to both xTop and the individual peptide precursors. Even when the protein abundance does not change across conditions, the strong dependence of iBAQ on the intensity of the top peptide precursor can lead to large amount of noise, such as in the case of YgeR (and RseA in Fig. 2C-G).

The two effects, the magnification of fold changes in protein abundances, and the additional noise, are simultaneously present when comparing different proteomes, which can obfuscate the biological interpretation of the data. We show below only one of these examples, but similar effects can be found in virtually every comparison in the manuscript.

In these plots we compared the proteome of *E. coli* cells grown on different growth media (same data as in Fig. 7A, with less genes highlighted to improve visibility); on the left, the

results obtained with xTop, while on the right the results obtained with iBAQ (in absence of any scaling). Several observations may be drawn:

- The group of *cus* proteins represent a striking example. Both absolute and relative quantification is affected by the choice of the proteomics method. In particular, while we observe all *cus* proteins to be expressed ~10-fold higher in the M9 medium compared to MOPS, we see a wide range of expression increases when using iBAQ, e.g. a ~100-fold increase for *cusA* and 10-fold increases for *cusS*. In this case, iBAQ would hint to a heterogeneity in the expression that is not observed with xTop.
- Similarly, the abundances of some σ^S -dependent proteins (yellow triangles) are observed to be strongly reduced in M9 media, which might lead to hypothesize a strong modulation of σ^S activity depending on the growth medium. Instead, the data generated with our workflow show only a moderate reduction in expression of these genes.
- Finally, more noise is present in the iBAQ case, especially for low-abundant proteins, which makes more difficult to determine which proteins are significantly up- or down-regulated.

Specific comments on xTop

• What is the connection between ϵ_p to the Equation N1.3? In the likelihood shown in N1.4, the random parameters seem to be x and σ , but it wasn't clear what the index r iterates over. Overall, the description of the likelihood and prior was very confusing, without proper introduction of notation and with constant change in subscripts across the paragraphs.

Appendix Note S1 (now Appendix Note S2) was indeed written in a confusing manner. Prompted by the Reviewer, we completely revised the note. It is now divided into two sections, the first one including an overview of the method and the second one discussing the detailed mathematical implementation.

The parameters to be optimized in the maximization of the posterior probability distribution are indeed x (vector including both log-transformed protein intensities and peptide efficiencies) and σ (including the parameters modelling the scatter in the log-transformed peptide intensities), the latter being incorporated in a larger vector s which also account for two of the three model parameters. The association between x , σ and the original quantities appearing in the model (intensities, efficiencies and scatter parameters) is now illustrated in detail in Figure N2.2.

• Also, it looks like the whole inference formulation is hierarchical Bayes - is that correct? How were the MAP estimates obtained? By sampling (e.g. Markov chain Monte Carlo?) or is there a closed-form solution? The prior elicitation process is described in page 15 and 16, but I found the entire writeup essentially anecdotal, lacking systematic evaluation through specification of multiple sets of priors and proper numerical evaluation.

The process is not hierarchical: there is only a prior term (for the parameters σ) and the likelihood (involving the parameters x , σ and the data y). The division between “ x ” and “ σ ” parameters is purely a matter of convenience, and there is no particular hierarchy between the two.

On MAP estimates and convergence. The MAP estimates are obtained by maximizing the posterior probability, i.e. minimizing its negative log (Eq. N2.12). The numerical technique is similar to gradient descent, except we alternatively set to zero the derivatives of Eq. N2.12 with respect to the two sets of parameters x and σ ; this is done alternatively using Eqn. N2.13 and N2.14, which provide (almost) closed forms for the parameters. As described in the Note, this process converges very quickly (about 15 iterations allow for a precision of 10^{-10} in the log-transformed posterior); the algorithm can be mathematically proven to converge to a minimum value (it is a monotonically decreasing sequence bounded from below).

However, similarly to gradient descent, the presence of multiple minima can lead the algorithm to convergence on local minima, rather than on a global optimum. Stimulated by the Reviewer question, we investigated whether that was the case, and found that in some rare cases the process did, in fact, converge on the wrong minimum, as described by Fig. N2.3. We have now changed the algorithm so that multiple minimizations are performed with different (randomly chosen) starting point for the parameters. This only affected a small fraction of the proteins (about 50).

On the prior. We essentially use only a single prior term, given in Eq. N2.7, which depends on a single parameter σ_{\min}^2 ; this parameter sets the minimum uncertainty associated to peptide precursors, as can be seen directly in Eq. N2.14. Such prior term has to be included in the posterior distribution to allow for the existence of a maximum (as you can see from Eq. N2.15, the posterior distribution is unbounded when $\sigma_{\min}^2 = 0$). Other prior terms are instead purely technical in nature, and can be avoided with a reformulation of the algorithm.

In this version of the Note we added Figure N2.4, which helps to provide an intuitive understanding of the action of the parameter σ_{\min}^2 . For small values of σ_{\min}^2 , xTop intensity is approximately proportional to the intensity of a single peptide precursor (purple line); for large values, all peptide precursors are weighted equally (yellow line). In our study, we set it to the value that minimizes the variance of the log-ratios, $\sigma_{\min}^2 \sim 10^{-2}$ (Fig. N2.5). However, the weak dependence of the results on σ_{\min}^2 implies that the choice of the prior does not affect strongly the final results of the method.

We apologize for the previous version of the Note, which was written in a confusing manner. We hope that the revised Note is more accessible.

• In Fig. N1.1, isn't the reason why xTop has the smallest variance because it is a weighted sum and each peptide's contribution to the whole sum is below 1 (ϵ_p), which in turn lowers the overall absolute level? You should probably switch the comparison to coefficient of variations, rather than variance.

The figure (now labeled Figure N2.5) indicates the variance of the log-transformed intensities, which is a measure of the scatter in intensity ratios, rather than in intensity differences, across the two replicates. Therefore, the absolute scale of the four different methods does not affect the analysis – gets canceled by the ratio. The quantity being plotted is now better indicated in the axis labels and in the caption.

• Figure 2C, main text line 200: I don't think the data shows any evidence of claiming superiority of xTop over others - in fact, they all look quite good. But I do welcome another measure comparable to iBAQ - it's obvious that iBAQ is not panacea to all problems. Would it be possible that xTop may have advantage specifically in the low abundance area. I would speculate that, since iBAQ adds up any peptide regardless of consistent detection across samples, xTop's weighting tactic should work better in low abundance proteins with patchy detection rates?

We agree that the data in Appendix Fig. S2D (previously Fig. 2C) is good for all methods ($r > 0.97$). However, the fact that the data spans about four orders of magnitude does not allow to appreciate finer details, which can only be appreciated by plotting data differently. This is done in Fig. S2B and S2E where we show the ratio of the intensities of the replicates, where a significant difference in the scatter is observed. Similarly, in the new version of the manuscript we radically changed the way we compared different proteomics methods, including comparisons of variance in log-ratios (Fig. 2H) or CVs (Fig. 2I-L) between replicates, or the quantification of proteins with large deviations in relative quantification as opposed to ribosome profiling (Fig. 3K).

To answer the latter question: Yes, xTop performs better than iBAQ when peptides are noisy or not consistently detected. This is clearly illustrated by the examples in Fig. 2C-G. However, note that a very large fraction of proteins were detected with only a few peptide precursors. For instance, in sample A1-1, 1910 proteins were detected, of which 765 were detected through three or less peptide precursors. For these proteins, which constitute 40% of the total proteins detected and the bulk of low-abundant proteins, it is crucial to limit the impact of noisy or missing peptides. See the answer below for a measure of the absolute quantification performance of iBAQ when only a few peptide precursors are detected.

• Another interesting thought: how is the overlap between every pair of the four approaches? Are they all well correlated? Can you provide a supplementary figure showing all pairwise plot? Such a plot would eventually reveal the area of abundance range where the four strategies differ.

We thank the reviewer for this suggestion. We have included such comparisons in Appendix Figure S5, using the same $N=1823$ proteins detected with all methods in all panels. As expected, xTop and TopPep1 are strongly correlated across the whole range of protein mass fractions. But keep in mind that TopPep1 detects about 100 fewer proteins. TopPep3 also shows a good correlation with xTop and TopPep1. In both cases, a group of low-abundant proteins arranged parallel to the diagonal but shifted downward. This feature corresponds to proteins that have been detected through the top peptide precursor only, in which case TopPep1 and TopPep3 intensities coincide, and hence the protein mass fractions are proportional to each other.

Finally, iBAQ is compared to the other three methods in the last row. For highly abundant proteins, all methods agree well with each other. However, for low abundant proteins (for which less peptide precursors are detected) iBAQ provides lower protein mass fractions compared to ribosome profiling, xTop or TopPep1. Indeed, the multivariate analysis in Appendix Note 3 revealed that proteins detected via only one peptide precursor are on

average 3-fold lower than ribosome profiling, and proteins detected via two peptide precursors are on average 1.7-fold lower, while such effect is not seen for xTop or TopPep1.

- Can your MATLAB code run on freely distributed software environment (like Octave)?

Unfortunately, it cannot, at least with the versions tested (Matlab R2015b and GNU Octave 4.2.2). However, it is an excellent suggestion, and we do regret that the code is not available to run in a free software. We are however considering to port xTop in Python in the context of a future project.

- What do you think would happen if you were to apply xTop to an eukaryotic proteome? How do you think it will fare against iBAQ? (Disclaimer: I am not a fan of iBAQ at all).

The origin of the proteome (e.g. prokaryote vs eukaryote) does not influence relative quantification or the impact of missing/noisy peptides, so xTop would still fare better than iBAQ from these points of view. The situation is more complicated for absolute quantification. In the case of eukaryotic cells, ribosome profiling rates might not reflect protein abundances, especially in cells that are slowly growing or non-proliferating. In this case, ribosome profiling does not provide absolute protein abundances that can be used to “calibrate” the xTop intensities.

However, if a “calibration” set of absolute protein abundances is not available, it might still be possible to use a different scaling factor for the xTop intensities. For instance, we found that rescaling the xTop intensities by a factor proportional to $L^{-0.6}$, where L is the protein length, allows to compensate for most of the biases observed in xTop and TopPep1 protein abundances. Another possibility is to use an iBAQ-like scaling factor, adding all peptide precursor efficiencies estimated with the xTop algorithm, and then dividing by the expected number of fully tryptic peptides. However, there might be other, unobserved, sources of bias. We plan to explore in depth these possibilities in a future work.

We briefly mention the possibility of different scaling factors in the Main Text, lines 289-292:

“The systematic biases in absolute quantification displayed above do not necessarily impact relative quantification. In particular, protein-specific biases such as that towards protein size can potentially be “fixed” by a condition-independent scaling factors (Appendix Note S3).”

And more extensively in Appendix Note S3, section “Bias removal via calibration of protein abundances”:

- Lastly, I'd like to enquire the behavior of xTop in the following scenario, for a protein with three peptides, with samples with identification and ion chromatogram extraction indicated as o's (detected) and x's (not detected) across five samples:

Peptide S1 S2 S3 S4 S5

1 0 0 0 x x

2 x 0 0 0 x

3 x x 0 0 0

To be concrete, we have generated an example corresponding to the detection pattern suggested by the Reviewer. Peptide intensities and the results for xTop are displayed in the figure below. Regarding absolute quantification, the xTop intensity is scaled to approximately match the intensity of the peptide with highest average intensity across the samples.

(1) Do you think we should quantify this protein at all?

It is hard to say without knowing the peptide intensities. The fact that the peptide precursors do not seem to be detected consistently across samples suggests we should not put much trust in the data, no matter which protein quantification method is used.

(2) Can ϵ_p be robustly estimated by xTop, with this particular configuration of detection (barring all limitations stemming from a small sample size)?

It can be estimated as long as the data allows to estimate the relative intensity between the peptides. In this case, the intensity ratio between peptides 1 and 2 can be inferred from samples S2 and S3, while the ratio between peptides 2 and 3 can be estimated from samples S3 and S4. Therefore, we were able to assign an xTop intensity in each of the five samples. It is of course much harder to assess whether the estimate is “robust”. The current implementation of xTop does not have a “robustness” metric (see answer below).

(3) The goal of the experiment was to compare abundance between group A (S1 + S2) and group B (S4 + S5). Is such comparison feasible?

Technically yes, as xTop provides estimates of the protein abundance in the four samples (red line in the plot above).

As you can already imagine, I am asking this question for one reason: I think you should consider incorporating a step where xTop declares certain proteins "not quantifiable". The criterion of retaining peptides detected in at least two samples is certainly not enough. The field of quantitative proteomics has always run towards one goal: quantifying more proteins. Almost no paper discusses the point of "quantifiable" fraction of proteome.

We feel that this is a fair assessment. The threshold of two samples per peptide is a purely technical one: with just one sample there is simply not enough data to estimate the uncertainty of the peptide intensity relative to that of the other peptides.

In this work, xTop is definitely geared towards maximizing the number of detected proteins. The number of minimal peptide precursors detected across samples can be easily tuned by the user of the MATLAB script; in our case, we set it to 3, just one more than the required minimum.

We will consider the problem of estimating uncertainties in both protein abundances and peptide efficiencies (and hence introduce "quality control" thresholds for the reported datasets) in a future work, together with that of improving the scaling of xTop to obtain bias-free absolute protein abundances without resorting to additional experimental data (e.g. ribosome profiling).

1. S. Hui *et al.*, Quantitative proteomic analysis reveals a simple strategy of global resource allocation in bacteria. *Molecular systems biology* **11**, 784 (2015).
2. M. Scott, C. W. Gunderson, E. M. Mateescu, Z. Zhang, T. Hwa, Interdependence of cell growth and gene expression: origins and consequences. *Science* **330**, 1099-1102 (2010).
3. D. H. de Groot, J. Hulshof, B. Teusink, F. J. Bruggeman, R. Planqué, Elementary Growth Modes provide a molecular description of cellular self-fabrication. *PLoS Comput Biol* **16**, e1007559 (2020).
4. H. Dourado, M. J. Lercher, An analytical theory of balanced cellular growth. *Nature communications* **11**, 1226 (2020).
5. E. J. O'Brien, J. A. Lerman, R. L. Chang, D. R. Hyde, B. Palsson, Genome-scale models of metabolism and gene expression extend and refine growth phenotype prediction. *Molecular systems biology* **9**, 693 (2013).
6. G. W. Li, D. Burkhardt, C. Gross, J. S. Weissman, Quantifying absolute protein synthesis rates reveals principles underlying allocation of cellular resources. *Cell* **157**, 624-635 (2014).
7. F. Mohammad, R. Green, A. R. Buskirk, A systematically-revised ribosome profiling method for bacteria reveals pauses at single-codon resolution. *eLife* **8**, (2019).

Thank you for sending us your revised manuscript. We have now heard back from the two reviewers who agreed to evaluate your revised study. As you will see below, they are both satisfied with the modifications made and they are supportive of publication. Reviewer #3 only lists a few minor remaining concerns which we would ask you to address in a minor revision.

Along the lines of the comment by reviewer #3, we would strongly encourage you to provide xTop implemented in an open source language.

We would also ask you to address the following editorial issues.

REFEREE REPORTS

Reviewer #1:

In the revision, the authors have addressed my previous concerns in adequate way. I found now the ms much better sound und understandable. In particular, I appreciate the effort the authors have put in emphasizing the biological significance of such rich data sets, which will help to reach out much broader audience. The finding about the 'composition of the catabolic proteins' and the allocation of proteome to specific catabolic needs for the carbon is quite intriguing.

Reviewer #3:

In the revised manuscript, the authors have addressed most major comments. Thank you for rewriting the notes - the parts on prior sensitivity analysis and MAP estimator are really helpful. The main text and the supplemental notes are much easier to understand now. I am happy to support the publication of the article. Here I list out remaining minor comments.

- Line 78: "crucial role of proteome cost" can be rephrased
- Line 179: "prioritizing the peptides which vary most consistently across conditions" - consistently in the sense that the levels are consistent within replicates of one growth condition? The sentence can be written again (sounded weird in the first pass)
- Line 215: Figure S2 does not show a "clear" improvement - I would somehow highlight this with boxes / ellipsis / arrows in the figure panel.
- I still believe that ribosome profiling is merely one way of benchmarking xTop results and it's not a perfect one since the technique captures actively translated proteins in general, rather than the abundance exactly proportional to the existing copy numbers in the cells or concentrations per cell (e.g. cytosolic fraction). At least I would acknowledge that this assumption is more applicable to the present study (E. coli proteome across growth conditions) and care has to be taken when xTop is applied to eukaryotic proteome analysis.
- Based on the description of xTop, I can't think of a reason why xTop cannot be implemented in an open-source language (R / python / C# or any variant) and provided to the research community. Otherwise, only people with Matlab license can apply the proposed method. If porting to python is an extremely time-consuming task, I understand the situation and trust that it will be facilitated in the future.

Editor comments

Along the lines of the comment by reviewer #3, we would strongly encourage you to provide xTop implemented in an open source language.

We agree with both the Reviewer and the Editor that an open-source implementation of the xTop method would be valuable for the community. For this reason, we have started porting the xTop code in Python. Unfortunately, translating the code between two such different programming languages is not trivial, and it will take some time before the code is fully written and tested. We have added a callout in the text to the fact that a Python version of the code is being developed and will be available at the same repository of the MATLAB code.

Reviewer #1:

In the revision, the authors have addressed my previous concerns in adequate way. I found now the ms much better sound und understandable. In particular, I appreciate the effort the authors have put in emphasizing the biological significance of such rich data sets, which will help to reach out much broader audience. The finding about the 'composition of the catabolic proteins' and the allocation of proteome to specific catabolic needs for the carbon is quite intriguing.

We thank the reviewer for the kind comments.

Reviewer #3:

In the revised manuscript, the authors have addressed most major comments. Thank you for rewriting the notes - the parts on prior sensitivity analysis and MAP estimator are really helpful. The main text and the supplemental notes are much easier to understand now. I am happy to support the publication of the article. Here I list out remaining minor comments.

- Line 78: "crucial role of proteome cost" can be rephrased

We substituted the sentence with the following (Lines 81-83):

“Quantitative data on protein abundances collected at this coarse-grained level across a spectrum of relevant growth conditions showed that the cost of protein synthesis is key to explain a number of ubiquitous microbial phenomena”

- Line 179: "prioritizing the peptides which vary most consistently across conditions" - consistently in the sense that the levels are consistent within replicates of one growth condition? The sentence can be written again (sounded weird in the first pass)

We modified the sentence as follows (lines 186-191):

"Importantly, the xTop protein intensity is obtained as a weighted average of all peptide precursor intensities. Peptides whose intensities display a large degree of mutual consistency across samples contribute the most to the intensity I_s^{xTop} , while peptides weakly correlated with the others contribute the least. Therefore, this method mitigates the impact of missing or noisy peptide precursors on the inferred protein intensities. An in-depth description of the method and of its implementation is provided in Appendix Note S2."

- Line 215: Figure S2 does not show a "clear" improvement - I would somehow highlight this with boxes / ellipsis / arrows in the figure panel.

We agree that the improvement is not visually clear in the figure; rather, the improvement is most clear in Fig. 2H where we summarize the variances across the replicates for each method. We have restructured the sentence (lines 224-226) so that it points more directly to Fig. 2H rather than to the Supplementary Text, as follows:

"However, a clear improvement of xTop over the other methods is seen when comparing the variance in the ratio of intensities between pairs of replicates, as summarized in Fig. 2H (see also Appendix Fig. S2B, S2E)."

- I still believe that ribosome profiling is merely one way of benchmarking xTop results and it's not a perfect one since the technique captures actively translated proteins in general, rather than the abundance exactly proportional to the existing copy numbers in the cells or concentrations per cell (e.g. cytosolic fraction). At least I would acknowledge that this assumption is more applicable to the present study (E. coli proteome across growth conditions) and care has to be taken when xTop is applied to eukaryotic proteome analysis.

We completely agree with the reviewer. We have added a cautionary statement in lines 341-346:

"For other conditions and organisms (e.g. slow-growing bacteria, or eukaryotic cells), the impact of protein degradation and protein translocation might not allow the use of ribosome profiling to calibrate relative protein abundances. In these cases, our versatile workflow can be adapted to use different quantities for the calibration of absolute protein abundances (see Appendix Note S3, section "Bias removal via calibration of protein abundances")."

- Based on the description of xTop, I can't think of a reason why xTop cannot be implemented in an open-source language (R / python / C# or any variant) and provided to the research community. Otherwise, only people with Matlab license can apply the proposed method. If porting to python is an

extremely time-consuming task, I understand the situation and trust that it will be facilitated in the future.

We agree with both the Reviewer and the Editor that an open-source implementation of the xTop method would be valuable for the community. For this reason, we have started porting the xTop code in Python. Unfortunately, translating the code between two such different programming languages is not trivial, and it will take some time before the code is fully written and tested. In the meantime, we have added a statement in the text pointing to the fact that a Python version of xTop is in development (Line 1027-1028):

“A Python version of the code is currently being developed, and will be made available at the same repository.”

Thank you again for sending us your revised manuscript. We are now satisfied with the modifications made and I am pleased to inform you that your paper has been accepted for publication.

Corresponding Author Name: Christina Ludwig

Manuscript Number: MSB-20-9536